# Fundamental limits of learning in sequence multi-index models and deep attention networks: high-dimensional asymptotics and sharp thresholds

Emanuele Troiani [1]   Hugo Cui [2]   Yatin Dandi [1][3]   Florent Krzakala [3]   Lenka Zdeborová [1]

## Abstract

In this manuscript, we study the learning of deep attention neural networks, defined as the composition of multiple self-attention layers, with tied and low-rank weights. We first establish a mapping of such models to sequence multi-index models, a generalization of the widely studied multi-index model to sequential covariates, for which we establish a number of general results. In the context of Bayes-optimal learning, in the limit of large dimension $D$ and proportionally large number of samples $N$, we derive a sharp asymptotic characterization of the optimal performance as well as the performance of the best-known polynomial-time algorithm for this setting –namely approximate message-passing–, and characterize sharp thresholds on the minimal sample complexity required for better-than-random prediction performance. Our analysis uncovers, in particular, how the different layers are learned sequentially. Finally, we discuss how this sequential learning can also be observed in a realistic setup.

## 1. Introduction

Recent years have witnessed a shift in paradigm in the automated learning from sequential data, such as language (Brown et al., 2020; Kenton & Toutanova, 2019). The backbone of many of these technological advances arguably lies in the use of *transformer architectures* (Vaswani et al., 2017) – a parametrization allowing the model to dynamically focus on relevant portions of the input, and extricate increasingly complex correlations between tokens through the successive application of *attention layers*. They

[1]Statistical Physics Of Computation Laboratory, École Polytechnique Fédérale de Lausanne (EPFL) [2]Center of Mathematical Sciences and Applications, Harvard University [3]Information Learning and Physics Laboratory, École Polytechnique Fédérale de Lausanne (EPFL). Correspondence to: Emanuele Troiani <emanuele.troiani@epfl.ch, lenka.zdeborova@epfl.ch>.

*Proceedings of the 42$^{nd}$ International Conference on Machine Learning*, Vancouver, Canada. PMLR 267, 2025. Copyright 2025 by the author(s).

thus constitute a function class able to represent intricate dependencies between the tokens of sequential covariates. In spite of their ubiquity, the study of such models from a theoretical viewpoint is still in its infancy.

This situation stands in contrast with the wealth of results on *multi-index models*, central to many theoretical works e.g. (Arous et al., 2021; Abbe et al., 2022; Veiga et al., 2022; Ba et al., 2022; Arnaboldi et al., 2023; Collins-Woodfin et al., 2024; Damian et al., 2024a; Bietti et al., 2023; Moniri et al., 2024; Berthier et al., 2024). Multi-index models define functions based on low-dimensional subspaces of covariates and are popular exemplars among theoreticians. However, shallow architectures like multi-index models are structurally simpler than attention models. Specifically, they (a) lack the hierarchical structure from multiple attention layers and (b) act on less structured, non-sequential covariates. These distinctions make it unclear whether results from multi-index models apply to multilayer attention architectures. Here, we show these models share a deep formal connection, enabling a transfer of insights and analytical approaches.

Namely, our first motivation is *to extend the theoretical framework of multi-index models to sequence models*. We consider the class of *sequence multi-index* (SMI) functions introduced by (Cui et al., 2024a; Cui, 2025), and defined over length $M$ sequences of $D$-dimensional tokens $\boldsymbol{x} \in \mathbb{R}^{D \times M}$ of the form

$$y_W^{\text{SMI}}(\boldsymbol{x}) = g\left(\frac{\boldsymbol{W}\boldsymbol{x}}{\sqrt{D}}\right). \tag{1}$$

where $\boldsymbol{W} \in \mathbb{R}^{P \times D}$ is a learnable projection matrix, and $g : \mathbb{R}^{P \times M} \to \mathbb{R}^K$ is a (possibly multi-dimensional) link function. The $K-$dimensional output could represent –to give an example– the predicted class probabilities in a sentiment classification task. As we shall see, SMI functions prove very versatile generalizations of multi-index models ($M = 1$) to sequence data of length $M > 1$, that can be studied using related ideas and theoretical tools.

Our second motivation is to deploy the formalism developed for SMI models to study complex neural architectures. Remarkably, the SMI class encompasses a range of models – with a particularly noteworthy example being *deep*

*attention* architectures. A deep attention model of depth $L$, $y_{\mathrm{DA}} : \mathbb{R}^{D \times M} \to \mathbb{R}^{M \times M}$, is defined recursively as

$$y_{\mathrm{DA}}(\boldsymbol{x}) = \sigma \left( \frac{\boldsymbol{x}_{L-1}^\top \boldsymbol{w}_L^\top \boldsymbol{w}_L \boldsymbol{x}_{L-1}}{D} \right), \qquad (2)$$

where $\forall l \in [\![1, L-1]\!]$,

$$\boldsymbol{x}_l = \boldsymbol{x}_{l-1} \left[ c \mathbb{1}_M + \sigma \left( \frac{\boldsymbol{x}_{l-1}^\top \boldsymbol{w}_l^\top \boldsymbol{w}_l \boldsymbol{x}_{l-1}}{D} \right) \right], \qquad (3)$$

with $\boldsymbol{x}_0 \equiv \boldsymbol{x} \in \mathbb{R}^{D \times M}$ and $\sigma : \mathbb{R}^{M \times M} \to \mathbb{R}^{M \times M}$ a non-linear map, taken in this work to be the softmax function. The deep attention function (2) corresponds to a composition of $L$ single-head self-attention layers (with tied key and query), specified by the strength $c \in \mathbb{R}$ of its skip connections, and tied key and query matrices $\boldsymbol{w}_l \in \mathbb{R}^{P_l \times D}$, for $l = 1, \dots, L$. We set the value matrix to identity. We show in Appendix A that the SMI class also describes even more generic multi-layer attention networks.

Before stating our main contributions, we elucidate the connection between the deep attention model (2) and the SMI model (1). While we focus on the single-head tied-weights architecture (2) for clarity, such a mapping can also be constructed for deep architectures with multiple heads, untied weights, and sequence-to-sequence models with outputs in $\mathbb{R}^{D \times M}$, as detailed in Appendix A. The connection between the SMIs and deep attention models constitutes a pathway towards many exciting research avenues in the study of neural architectures beyond the exemplar of multi-layer perceptrons –such as classical multi-index models–, while allowing the transfer of ideas from the study of the latter.

**Mapping deep attention to the SMI model–** Starting from a deep attention model (2) with weights $\{\boldsymbol{w}_l^\star\}_{l=1}^L$, let us introduce the projections (a.k.a the eponymous indices) $\{Z_l \in \mathbb{R}^{P_l \times M}\}_{l=1}^L$ of the input data $\boldsymbol{x} \in \mathbb{R}^{D \times M}$ on each weight matrix $\boldsymbol{w}_l^\star \in \mathbb{R}^{P_l \times D}$: $Z_l = \boldsymbol{w}_l^\star \boldsymbol{x} / \sqrt{D}$. Layer by layer, the post-activation sequences $\boldsymbol{x}_l$ build on the initial sequence $\boldsymbol{x}$ by adding increasingly sophisticated correlations between tokens. A key observation, however, is that successive post-activations $\boldsymbol{x}_l$ retain a simple formal expression, and depend non-linearly on the input $\boldsymbol{x}$ only through the indices $Z_1, \dots, Z_l$. More precisely, at every layer $l$, the post activation $\boldsymbol{x}_l$ can be written as

$$\boldsymbol{x}_l = \boldsymbol{x} B_c^l(Z_1, \dots, Z_l), \qquad (4)$$

where we introduced a function $B_c^l : \mathbb{R}^{P_1 \times M} \times \cdots \times \mathbb{R}^{P_l \times M} \to \mathbb{R}^{M \times M}$, subsuming the complex inter-token interactions resulting from the propagation of the input through previous layers. The sequence of mixing functions

$\{B_c^l\}_{l=1}^L$ are defined recursively as:

$$
\begin{aligned}
B_c^l&(Z_1, \dots, Z_l) = \\
&B_c^{l-1}(Z_1, \dots, Z_{l-1})^{(1-\delta_{l,L})} \Big[ (1 - \delta_{l,L}) c \mathbb{1}_M \\
&+ \sigma \left( B_c^{l-1}(Z_1, \dots, Z_{l-1})^\top Z_l^\top Z_l B_c^{l-1}(Z_1, \dots, Z_{l-1}) \right) \Big],
\end{aligned} \qquad (5)
$$

with $B_c^0 = \mathbb{1}_M$, the identity matrix. The claim (4) then follows from the definitions (5) and (2) by recursion, finally yielding

$$y_{\mathrm{DA}}(\boldsymbol{x}) = B_c^L(Z_1, \dots, Z_L). \qquad (6)$$

Eq. (6) can be written more compactly and evocatively as

$$y_{\mathrm{DA}}(x) = g \left( \frac{\boldsymbol{W}^\star \boldsymbol{x}}{\sqrt{D}} \right), \qquad (7)$$

introducing the total weights $\boldsymbol{W}^\star \in \mathbb{R}^{P \times D}$, defined as the vertical concatenation of the weight matrices $\{\boldsymbol{w}_l^\star\}_{l=1}^L$ along their first dimension, and denoting $P = P_1 + \dots + P_L$. The link function $g$ in (7) follows from a redefinition of $B_c^L$, as can be read from (6), and takes values in $\mathbb{R}^{M \times M} \approx \mathbb{R}^K$ for $K = M^2$. One then observes that the equivalent model (7) is an SMI model (1), whose width $P$ is related to the depth of the original deep attention model (2). The recursive structure of the deep network (2) is encoded in the way the highly structured function $g$ acts on the different rows of its argument $\boldsymbol{W}^\star \boldsymbol{x} \in \mathbb{R}^{P \times M}$. Importantly, the $P$ rows generically do not play symmetric roles, reflecting the structure of the original deep attention function.

**Main contributions –** Building on the above mapping, we study the statistical and computational limits of learnability from data generated by the models (1) and (2). Namely, we consider training data composed of $N$ samples of Gaussian i.i.d. input sequences, and output labels generated by the SMI model (1) with random i.i.d. Gaussian weights (the same realization of weights for every sample). In the limit of large dimension $D$ and number of samples $N$ with $N/D, P, M, K = \Theta(1)$, we characterize the minimal prediction error achievable information-theoretically and also by a class of algorithms conjectured optimal among all polynomial algorithms. This is achieved by the following technical contributions:

• We show how to rigorously generalize results previously established for multi-index models to the SMI models – including the notions of phase transition for weak recovery, and optimal message-passing algorithms (Damian et al., 2024b; Troiani et al., 2025).
• We derive sharp asymptotic expressions for optimal estimation errors achievable information theoretically and with the Approximate Message-Passing (AMP) algorithm, in the limit of large covariate dimension $D$, and proportionally

large number of samples $N$, but finite sequence length $M$, weight rank $P$ and depth $L$. This characterization builds on the analyses of (Donoho et al., 2009; Javanmard & Montanari, 2013; Gerbelot & Berthier, 2023) for AMP, which is optimal among first-order methods (Celentano et al., 2020).
• We extend the numerical evaluation and analysis of the resulting asymptotic equations from $M = 1$ in (Aubin et al., 2018; Troiani et al., 2025), and $P = 1$ in (Cui et al., 2024a; Cui, 2025) to SMI models with both sequence length $M > 1$, and number of indices $P > 1$.

Equipped with the above theoretical results, we extract the *weak recovery thresholds*, namely the sample complexity required for a better-than-random estimation, for the weights of every layer in the case of a two-layer attention model. We show that different layers are learnable at different sample complexities, implying sequential learning of the different weights. We then briefly discuss how such layer-wise learning also arises in more realistic settings.

### Further related works

**Analyses of single attention layers –** Many theoretical studies of attention-based models (Vaswani et al., 2017) consider the case of a *single* attention layer, studying it in isolation (Lopardo et al., 2024; Zhang et al., 2024; Li et al., 2023b; Tian et al., 2023; Jelassi et al., 2022). This rich line of works shed light on various aspects of these models, such as their inductive bias (Sahiner et al., 2022; Ataee Tarzanagh et al., 2023; Tarzanagh et al., 2023), training dynamics (Li et al., 2023a), or expressivity (Fu et al., 2024). A fraction of this body of works was devoted to characterizing attention models in the limit of large dimensions. (Lu et al., 2024) and (Rende et al., 2024) provide tight asymptotic characterization of the error achieved by linear attention layers, respectively for in-context learning and next-token prediction. (Cui et al., 2024a) uncover a phase transition between semantic and positional learning in a model of attention mechanism with low-rank weights. For a related model, (Marion et al., 2024) study its population gradient dynamics, and prove its asymptotic Bayes-optimality for a regression task. Because these works focus on single attention layers however, they cannot capture the compound effect of multiple successive attention layers in building increasingly complex correlations between tokens present in multilayer models such as (2).

**Analyses of deep attention models –** In an effort to supersede those limitations, a recent stream of works has striven to tackle the problem of multiple attention layers, under varying simplifications. (Geshkovski et al., 2024; 2023) study the propagation of a signal through multiple attention layers with frozen weights. (Ahn et al., 2023; Von Oswald et al., 2023) describe how the successive layers can implement gradient descent steps, establishing how an optimization algorithm can be encoded by such an architecture. Deep attention models have also been studied through the lens of their capacity (Edelman et al., 2022) and expressivity (Hahn, 2020), and scaling limits (Bordelon et al., 2024). The training dynamics of deep attention models were ascertained close to initialization by (Bietti et al., 2023), and fully by (Abbe et al., 2024a), in exchange of the assumption of diagonal weights. Closer to our setting, (Tiberi et al., 2024) study the Bayesian learning of a multilayer attention, assuming trainable value matrices but frozen key and query matrices. In this respect, the model is complementary to the one considered in the present work, where in contrast the learning of key and query weights is addressed, while value matrices are fixed to identity. Another key difference lies in the fact that the attention matrix at every layer is computed using the bare input $x$ in (Tiberi et al., 2024), as opposed to the output of the previous layer $x_{l-1}$ as in (2).

**Learning multi-index models –** Multi-index models correspond to target function of the form $y(x) = g(Wx)$, specified by a link function $g$ and a finite rank matrix $W \in \mathbb{R}^{P \times D}$, with $P = \Theta(1)$, and can be viewed as two-layer MLPs with weights $W$ and activation $g$. They provide a natural exemplar for functions that operate on high-dimensional covariates, but only depend on a finite number of directions. Several authors have used multi-index models as testbeds to explore the ability of neural networks to learn low-dimensional subspaces in high dimensions (Ba et al., 2022; Moniri et al., 2024; Dandi et al., 2023; 2025; Cui et al., 2024b). A substantial stream of works has furthermore been devoted to studying the behavior of (stochastic) gradient descent on such non-convex objectives (Saad & Solla, 1995; 1996; Arous et al., 2021; Bietti et al., 2023; Simsek et al., 2025; Barak et al., 2022; Berthier et al., 2024; Abbe et al., 2023; Damian et al., 2022; Glasgow, 2023; Arnaboldi et al., 2024b), with the work of (Oko et al., 2024) providing insights for more generic scalings of the number of hidden units $P$ with the dimension $D$. In closer relation to the present work, (Troiani et al., 2025) ascertain the fundamental computational limits of learning multi-index functions, leveraging an analysis of AMP algorithms. In this manuscript, we establish a deep connection between deep attention models (2) and SMI models (1), a generalization of multi-index models introduced by (Cui et al., 2024a; Cui, 2025), allowing to borrow from the aforementioned wealth of insights and results. The connection to deep attention models was not noticed in (Cui et al., 2024a; Cui, 2025), and (Cui, 2025) studied the empirical risk minimization for the SMI models using the replica method, and has not established the fundamental statistical and computational limits for the SMI function class.

**Statistical and computational limits of learning –** The optimal prediction errors associated with single-layer neural network functions were first derived in a line of foundational works (Gardner & Derrida, 1988; Sompolinsky et al., 1990; Györgyi, 1990) for perceptrons, and rigorously in (Barbier et al., 2019). The statistical limits of learning in committee machines were explored in (Schwarze & Hertz, 1992; Schwarze, 1993; Monasson & Zecchina, 1995), and its computational limits in (Aubin et al., 2018; Troiani et al., 2025). The case of large-width networks was covered in (Zavatone-Veth et al., 2022) for linear models, and (Cui et al., 2023; Camilli et al., 2023; Maillard et al., 2024) for non-linear models. Closer to our work, (Erba et al., 2024) consider a related model of bilinear sequence regression, characterizing its optimal errors, and proposing an associated AMP algorithm matching these performances. (Abbe et al., 2024b) study the weak learnability of a variety of tasks for transformers trained with descent algorithms.

## 2. Fundamentals limits of learning SMI models

The question of determining the statistical and computational limits of learning has traditionally been thoroughly explored for multi-index models (Gardner & Derrida, 1988; Schwarze & Hertz, 1992; Barbier et al., 2019; Aubin et al., 2018; Troiani et al., 2025). In this first section, we show how these ideas can be extended to SMI models (1).

### 2.1. Bayes-optimal learning of SMI models

We are interested in the statistical inference problem of estimating the weights $W^\star$ of an SMI model $y_{W^\star}^{\mathrm{SMI}} = g\left(W^\star x / \sqrt{D}\right)$ from $N$ observations $\mathcal{D} \equiv \{(x^\mu, y_{W^\star}^{\mathrm{SMI}}(x^\mu))\}_{\mu=1}^N$ of the function on identically and idependently sampled (i.i.d) covariates $x^\mu \in \mathbb{R}^{D \times M}$ with i.i.d standard Gaussian components. We further consider a random instance of the SMI model, where the weights $W^\star$ have been randomly sampled from a prior distribution, which we assume for definiteness to be i.i.d Gaussian over all the components. For this problem, we ask the question of the *optimal statistical and computational* prediction error that can be deduced from the estimation of the weights $W^\star$ from the observations $\mathcal{D}$: what is the information-theoretically lowest reconstruction error? What is the lowest error achievable by polynomial-time algorithms?

We answer these questions in the *Bayes-optimal setting*, in which the statistician has full knowledge of the parameters of the model except the realization of its weights $W^*$. The estimator $y(\cdot)$ minimizing the prediction error $\mathbb{E}_x \|y(x) - y_{W^\star}^{\mathrm{SMI}}(x)\|^2$ is then given by the mean $y(x) =$

$\mathbb{E}_W[y_W^{\mathrm{SMI}}(x)|\mathcal{D}]$ over the Bayesian posterior distribution

$$\mathbb{P}(W|\mathcal{D}) \propto e^{-\frac{1}{2}\mathrm{Tr}[WW^\top]} \prod_{\mu=1}^N \delta\left(y_{W^\star}^{\mathrm{SMI}}(x^\mu) - y_W^{\mathrm{SMI}}(x^\mu)\right). \tag{8}$$

**Asymptotic limit –** In the limit of large covariate dimension $D$ and large number of samples $N$, sampling the posterior distribution (8) is generically computationally challenging. It is on the other hand possible, in this limit, to access exact theoretical characterizations of key statistics of the optimal estimators. We detail this analysis in the following sections. More precisely, we consider the asymptotic limit of $D, N \to \infty$ while $\alpha \equiv N/D = \Theta(1)$. We further assume that the sequence length $M$, model width $P$ and output dimension $K$ remain finite : $M, P, K = \Theta(1)$.

**Methodology –** The following technical results are enabled by the key observation that the SMI model (1) can further be viewed as an ordinary multi-index model, provided the data and weights are properly reshaped. More precisely, to go from an SMI model to an equivalent multi-index model, one can flatten the input $x \in \mathbb{R}^{D \times M}$, viewed as a length $M$ sequence of $D$-dimensional tokens, into a vector in $\mathbb{R}^{MD}$. In parallel, the matrix of weights $W^\star \in \mathbb{R}^{P \times D}$ should be mapped into a block-diagonal matrix of size $PM \times DM$ with each of the $M$ blocks on the diagonal being equal to $W^\star$, as detailed in Appendix B. In words, an SMI model can be viewed as a multi-index model acting on flattened sequences, with a non-separable prior on the parameters.

This mapping allows us to borrow from previous results on multi-index models. On the one hand, the computational limits can be determined along the lines of (Troiani et al., 2025), leveraging an analysis of the generalized-AMP (GAMP) algorithm, which is provably optimal within the class of first-order methods (Celentano et al., 2020). This analysis can be carried out using methods developed in (Gerbelot & Berthier, 2023) on the state evolution of GAMP. The statistical limits of learning may be, on the other hand, obtained by leveraging the rigorous results on the free-energy of the posterior measure (8) for multi-index models established in (Aubin et al., 2018), as sketched in Appendix B.2, or using the non-rigorous replica method from statistical physics, as detailed in Appendix D.5.

### 2.2. Statistical limits: Bayes-optimal error

To state our main results, we start by defining a finite-dimensional *output channel* induced by Eq. (1) that will be referred to throughout the subsequent sections:

$$Y(\omega, V) = g\left(\omega + \sqrt{V}Z\right), \tag{9}$$

where $Z \in \mathbb{R}^{P \times M}$ and $Z_{ij} \stackrel{\text{i.i.d.}}{\sim} \mathcal{N}(0, 1)$. On an intuitive level, $\omega \in \mathbb{R}^{P \times M}$ and $V \in \mathbb{S}_P^+$ (a positive semidefinite matrix od size $P$) correspond to estimates of the mean and covariance of the inputs to the nonlinearity $g(\cdot)$ in Eq. (1).

With the above definition and assumptions on $g$ specified in Appendix B.2, we are in a position to state our first result, characterizing the asymptotic Bayes-optimal error for a general SMI model of the form (1):

**Theorem 2.1.** *Consider the SMI model (1) with nonlinearity g. Let $\xi \in \mathbb{R}^{P \times M}$ denote a matrix with entries $\xi_{ij} \stackrel{\text{i.i.d.}}{\sim} \mathcal{N}(0, 1)$. Let $H_Y(Q)$, denote the conditional entropy of the associated output channel $Y$ defined by Eq. (9) with $\omega = \sqrt{Q}\xi$ and $V = \mathbb{1}_P - Q$ for the so-called overlap $Q \in \mathbb{S}_P^+$. Suppose that the following $\sup \inf$ problem:*

$$
\sup_{\hat{Q} \in \mathbb{S}_P^+} \inf_{Q \in \mathbb{S}_P^+} \left\{ -\tfrac{1}{2}\mathrm{Tr}(Q\,\hat{Q}) - \tfrac{1}{2}\log(\mathbb{1}_P + \hat{Q}) \right.
$$
$$
\left. + \tfrac{1}{2}\hat{Q} + \alpha H_Y(Q) \right\}. \tag{10}
$$

*admits a unique global extremizer $\hat{Q}^\star, Q^\star$. Then the asymptotic Bayes-optimal prediction error is given by:*

$$
\mathbb{E}_{\boldsymbol{x}} \left\| y_{\boldsymbol{W}^\star}^{\text{SMI}}(\boldsymbol{x}) - \mathbb{E}_{\boldsymbol{W}} \left[ y_{\boldsymbol{W}}^{\text{SMI}}(\boldsymbol{x}) | \mathcal{D} \right] \right\|^2 \xrightarrow{D \to \infty} \tag{11}
$$
$$
\mathbb{E}\left[ \|g(\sqrt{Q^\star}\xi)^2\|^2 - \left\langle g(\xi), g\left(\sqrt{\mathbb{1}_P - Q^\star}Z + \sqrt{Q^\star}\xi\right) \right\rangle \right]
$$

*and the Bayes-optimal estimation error is $\|\boldsymbol{W}^*(\boldsymbol{W}^*)^\top - \mathbb{E}_{\boldsymbol{W}}\left[\boldsymbol{W}\boldsymbol{W}^\top | \mathcal{D}\right]\|^2 \to 1 - \|Q^*\|_2^2$.*

The proof of Theorem 2.1 is outlined in Appendix B. A complementary non-rigorous derivation using the replica method is provided in Appendix D.5.

### 2.3. Computational Limits: Weak-recovery thresholds

We next move to presenting a sharp asymptotic characterization of the computational limits of the Bayesian estimation problem (8). Our approach relies on the study of a GAMP algorithm (Donoho et al., 2009), reported in Algorithm 1, building on two of its remarkable properties. First, the GAMP algorithm associated to a Bayes-optimal estimation problem is provably optimal among first-order methods (Celentano et al., 2020) and runs in polynomial time. Consequently, the sample complexity required for the learnability for GAMP implies a computational lower bound on the class of all first-order algorithms, which includes in particular widely used gradient descent methods. Secondly, the performance of GAMP algorithms admits a sharp asymptotic description in terms of finite-dimensional variables. This description, known as the *state evolution* equations (Bayati & Montanari, 2011; Javanmard & Montanari, 2013), allow the analysis of GAMP in the high-dimensional limit $D \to \infty$. The state evolution description is formalized through the following Lemma:

---

**Algorithm 1** GAMP for SMI model

**Inputs** : $\mathcal{D} = \{\boldsymbol{x}^\mu, y^\mu\}_{\mu=1}^N$
**Initialize** $\hat{\boldsymbol{W}}_i^0 = \mathcal{N}(0, \mathbb{1}_P), \hat{C}^0 = \mathbb{1}_P, g_\mu^0 = 0_{P \times M}$
**for** $t \leq t_{\max}$ **do**
$\quad V^t = \hat{C}^t \in \mathbb{R}^{P \times P}$
$\quad \omega_{m,\mu}^t = \frac{1}{\sqrt{D}}\sum_{i=1}^D \boldsymbol{x}_{m,i}^\mu \hat{\boldsymbol{W}}_i^t - V^t g_{m,\mu}^t \in \mathbb{R}^{P \times M}$
$\quad g_{m,\mu}^t = \left[g_{\text{out}}(y^\mu, \omega_\mu^t, V^t)\right]_m \in \mathbb{R}^P$
$\quad A^t = -\frac{\alpha}{N}\sum_{\mu,m=1}^{N,M} \left[\partial_\omega g_{\text{out}}(y^\mu, \omega_\mu^t, V^t)\right]_{mm} \in \mathbb{R}^{P \times P}$
$\quad b_i^t = \frac{1}{\sqrt{D}}\sum_{\mu,m=1}^{N,M} \boldsymbol{x}_{mi}^\mu g_{m,\mu}^t + A^t \hat{\boldsymbol{W}}_i^t \in \mathbb{R}^P$
$\quad \hat{\boldsymbol{W}}_i^{t+1} = (\mathbb{1}_P + A^t)^{-1}b_i^t$
$\quad \hat{C}^{t+1} = (\mathbb{1}_P + A^t)^{-1}$
**end for**

---

**Lemma 2.2** (State evolution (Gerbelot & Berthier, 2023)). *Define $g_{\text{out}} : \mathbb{R}^{M \times M} \to \mathbb{R}^{P \times M}$ to be the following "denoiser":*

$$
g_{\text{out}}(Y, \omega, V) := \mathbb{E}[Z \,|\, Y], \tag{12}
$$

*where the conditional expectation is w.r.t the output channel in Eq. (9). Let $\hat{\boldsymbol{W}}^t$ denote the iterates of Algorithm 1 at time $t \in \mathbb{N}$. Assuming $g_{\text{out}} \in \mathcal{C}^2$, under the high-dimensional limit $N, D \to \infty$ with fixed ratio $\alpha = {}^N\!/_D$, constant $M$ and any finite time $t$, the limiting overlaps converge in probability to the overlap $Q^t$:*

$$
{}^1\!/_d \hat{\boldsymbol{W}}^t \hat{\boldsymbol{W}}^{t\top} \xrightarrow{P} Q^t, \quad {}^1\!/_d \hat{\boldsymbol{W}}^t \boldsymbol{W}^{\star\top} \xrightarrow{P} Q^t, \tag{13}
$$

*with $Q^t$ satisfying the state evolution equations*

$$
Q^t = F\left(\alpha \, \mathbb{E}_{Y,\xi}\left[g_{\text{out}}(Y, \sqrt{Q^{t-1}}\xi, \mathbb{1}_P - Q^{t-1})^{\otimes 2}\right]\right), \tag{14}
$$

*from an initial condition $Q^0$. In (14), we denoted $F(\hat{Q}) = (\mathbb{1}_P + \hat{Q})^{-1}\hat{Q}$, and employed the shorthand $Y = Y(\sqrt{Q}\xi, \sqrt{\mathbb{1}_P - Q})$, with $\xi \in \mathbb{R}^{P \times M}$ denoting a matrix with entries $\xi_{ij} \stackrel{\text{i.i.d.}}{\sim} \mathcal{N}(0, 1)$.*

In Appendix D.2, we establish that the fixed points of the state evolution equations (14) are in one-to-one correspondence with the stationary points of the objective function defined in Equation (10). Consequently, in cases where Equation (10) admits a unique non-zero stationary point $Q^\star \in \mathcal{S}_p^+$ corresponding to the global extremum, the GAMP algorithm, provided it converges to a non-zero limit, necessarily attains an overlap ${}^1\!/_d\hat{\boldsymbol{W}}^t\hat{\boldsymbol{W}}^{t\top}$ converging to $Q^\star$. It then follows from Equation 11 that GAMP asymptotically achieves Bayes-optimal performance. The proof of Lemma 2.2, which builds upon (Berthier et al., 2020), is outlined in

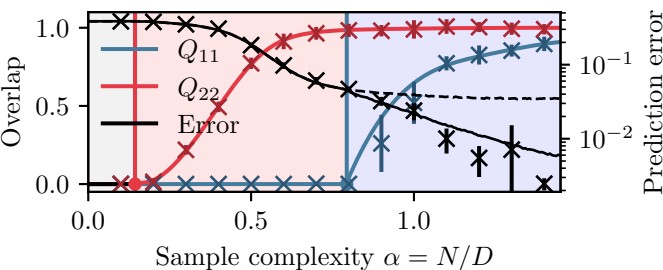
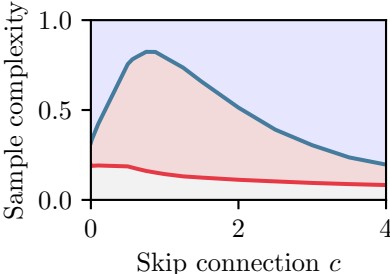

*Figure 1.* (**Left**) Diagonal elements of the overlap $Q$ (13) (red and blue) and prediction error (black) achieved by GAMP, as a function of the sample complexity $\alpha = {}^N/_D$, for a two-layer attention model $P_1 = P_2 = 1, c = 1, M = 2$. Off-diagonal elements of $Q$ are zero, and thus not plotted. Crosses: numerical implementations of GAMP in dimension $D = 1000$, averaged over 16 runs. Convergence was reached with at most 50 iterations. Continuous lines: theoretical prediction from Eqs. 14 and (11). Dashed line: prediction error of GAMP when the first layer weights are fixed to zero. (**Right**) Weak recovery thresholds $\alpha_1$ (in red) and $\alpha_2$ (in blue), above which the second and first layer weights are respectively retrieved, as a function of the skip connection strength $c$. The red (resp. blue) lines indicate the sample complexities above which the second (resp. first) layers can be learned.

Appendix B. We also provide a complementary derivation of this result from the Belief Propagation scheme (Pearl, 1988) associated with the inference problem (8) in Appendix D. Lemma 2.2 provides an asymptotic description of the performance of GAMP (Algorithm 1), which can be leveraged to ascertain when reconstruction is computationally achievable, what sample complexity is required to do so, and whether the algorithm achieves the Bayes-optimal performance.

Hereafter, let us consider more particularly the case of even SMI models, namely those invariant under the reflection $\mathbf{W} \leftrightarrow -\mathbf{W}$ of their weights. This is the case, for instance, of the tied deep attention model (2). As a consequence of this symmetry, $Q = 0$ is a trivial fixed point of the state-evolution of GAMP, and the latter fails to reconstruct any part of the row-space of $W^\star$. As is well known for single and multi-index models (Mondelli & Montanari, 2018; Troiani et al., 2025), the breaking of such a symmetry necessitates the introduction of *side-information*, defined as an additional observation $\lambda W^\star + \sqrt{1-\lambda}\xi$, with $\xi \in \mathbb{R}^{P \times D}$ a matrix with Gaussian i.i.d entries, and $\lambda \in \mathbb{R}$. The GAMP algorithm in the presence of side information differs from Algorithm 1 only in the updates of $\hat{\mathbf{W}}, \hat{C}$, as we discuss in Appendix D. The introduction of side-information allows us to define a number of *weak-recovery thresholds*. Given a subspace $U \subseteq \mathbb{R}^p$ the weak recovery threshold for $U$ is defined as the smallest $\alpha$ such that the iterates $\hat{\mathbf{W}}$ of the GAMP algorithm when projected along the rows of $W^\star$ achieve non-vanishing components along all directions of $U$, with arbitrarily small side-information coefficient $\lambda$. We discuss the formal definitions and existence of such thresholds in Appendix C, based on the analysis in (Troiani et al., 2025). A special threshold we can characterize in closed form is the *weak-recovery threshold for initial recovery*, that is the smallest sample complexity $\alpha_1$ such that GAMP can

achieve non vanishing overlap in *any* subspace $U$ without having any significant overlap at all. This thresholds can be characterized in closed-form in terms of the stability of the fixed-points of the state-evolution (14) of Lemma 2.2. Indeed, the iterates $\hat{W}^t$ can escape the neighborhood of the fixed point $Q = 0$ in the presence of vanishing side-information $\lambda$ if and only if the fixed-point $Q = 0$ becomes unstable. The weak-recovery threshold for initial recovery can thus be determined by performing a linear stability analysis. Notice that once this first subspace $U_1$ is recovered, then a new threshold $\alpha_2$ emerges: $\alpha_2$ describes the minimal sample complexity such that the fixed point becomes unstable to perturbations orthogonal to $U_1$. This procedure can be then repeated, creating a natural ordering in the learning of $\text{row span}(W^\star)$ as a function of the sample complexity, which we call *grand-staircase* mechanism in analogy with Definition 6 in (Troiani et al., 2025) . This discussion, which we prolong in Appendix C, is formalized in the following theorem.

**Theorem 2.3.** *Suppose that $g_{\text{out}} \in \mathcal{C}^2$ with $G := \partial_\omega g_{\text{out}}(Y, 0, \mathbb{1}_P) \in \mathbb{R}^{P \times P \times M \times M}$ denoting the tensor-Jacobian of $g_{\text{out}}$ with respect to $\omega$. Define $\mathcal{F}(\mathcal{X})$ to be the following linear operator on $\mathbb{R}^{P \times P}$*

$$\mathcal{F}(\mathcal{X})_{ij} = \sum_{a,b=1}^{M} \sum_{kl=1}^{P} \mathbb{E}_Y \left[ G_{ikab} \mathcal{X}_{kl} G_{ljab} \right], \quad (15)$$

*where $Y$ is distributed as in (9). Then the weak recovery threshold for initial recovery – namely for reconstructing any subspace – is*

$$\frac{1}{\alpha_1} = \sup_{\mathcal{X} \in \mathbb{S}_+^{Pl}, \|\mathcal{X}\|_F = 1} \|\mathcal{F}(\mathcal{X})\| \quad (16)$$

*where $\|\mathcal{X}\|_F$ is the Frobenius norm and $\mathcal{S}_+^P$ is the set of positive semi-definite matrices of size $P \times P$. More generally,*

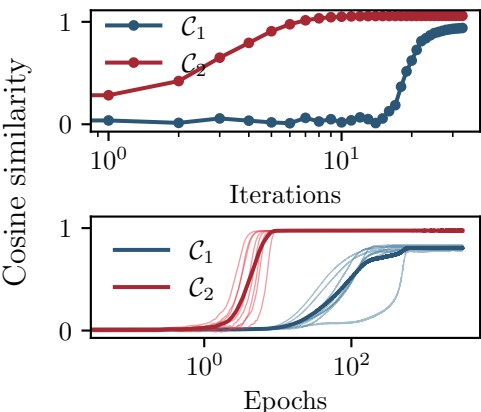

*Figure 2.* (**Top**) Evolution of the performance of GAMP with the number of iteration, as measured by the cosine similarity $\mathcal{C}_l = |(\boldsymbol{w}_l^\star)\hat{\boldsymbol{w}}_l^\top|/\|\hat{\boldsymbol{w}}_l\|$ between the GAMP estimate $\hat{\boldsymbol{w}}_l$ of the $l$−th layer weights and the target weights $\boldsymbol{w}^\star$, for $L = 2, P_1 = P_2 = 1, c = 1, M = 2$. We display a single run of the algorithm in dimension $D = 1000$ and sample complexity $\alpha = N/D = 1.2$ starting from the first iteration. (**Bottom**) Evolution of the cosine similarity, for the same target function, when training the same model using SGD. We display 8 runs of the algorithm in dimension $D = 500$ and sample complexity $\alpha = 15$, with the average indicated in bold. The numerical experiments were performed at $\lambda = 1.4 \times 10^{-4}, \eta = 15$, and batch size 200, with each batch used for 3 consecutive iterations.

*for a subspace $U \subseteq \mathbb{R}^p$, we denote by $\alpha_U$, the threshold for weak-recovery of all directions in $U$, defined as:*

$$\alpha_U = \inf\{\alpha : \lim_{\lambda \to 0^+} \lim_{t \to \infty} P_U Q_\lambda^t (P_U)^\top \succ 0\}, \quad (17)$$

*where $P_U$ denotes the projection onto $U$ and $Q_\lambda^t \in \mathbb{R}^{P \times P}$ are recursively defined by:*

$$Q_\lambda^{t+1} = F_\lambda \left( \alpha \, \mathbb{E}_{Y,\xi} \left[ g_{\text{out}}(Y, \sqrt{Q_\lambda^t}\xi, \mathbb{1}_P - Q_\lambda^t)^{\otimes 2} \right] \right), \quad (18)$$

*with $F_\lambda(\hat{Q}) = (\hat{Q}(1 - \lambda) + \lambda\mathbb{1}_P)(\mathbb{1}_P + \hat{Q}(1 - \lambda))^{-1}$.*

*Definition 17 then specifies an ordering of subspaces $U_1 \subset U_2 \subset ...U_k \subseteq \mathbb{R}^k$ such that $\alpha_{U_1} < \alpha_{U_2} < ...\alpha_{U_k}$. We denote $\alpha_{U_j}$ for $j = 1, 2, 3...$ in the above ordering as $\alpha_j$, with $\alpha_1$ matching the description in Equation 9, and in general use the shorthand $\alpha_k = \alpha_{U_k}$.*

The existence of the limit defined by Eq. (17) is a consequence of the monotonicity of the asymptotic dynamics of GAMP described by Eq. (18), see Appendix B.

# 3. Results for deep attention models

The previous section delineates the fundamental limits of learning in the class of SMI models (1). We now show how

these results seamlessly transfer to deep attention models (2), and discuss in particular the consequences of Theorem 2.3 for these architectures.

### 3.1. Layer-wise learning in depth $L = 2$ attention

We examine for definiteness the case of a two-layer attention model (2) ($L = 2, M = 2, P_1 = P_2 = 1, c = 1$). Fig. 1 (left) illustrates the theoretical predictions of Lemma 2.2 for the prediction error (11) and overlap $Q$ (13) (right) achieved by GAMP, as a function of the sample complexity $\alpha$. Before discussing the curves, let us highlight two remarkable features of the behavior of GAMP in this model. First, we observe that the values of these metrics at convergence remain unchanged if GAMP is initialized with strong side information. This finding is a telltale sign of the absence of computational-to-statistical gaps for this particular problem, and strongly suggests that GAMP is in this case Bayes-optimal. Secondly, we notice that the off-diagonal elements $Q_{12}, Q_{21}$ of the overlap $Q$ (13) vanish at convergence.

A striking observation from Fig. 1 is that the learning proceeds *in sequential steps*, as the sample complexity $\alpha$ is increased. For $\alpha < \alpha_1 \approx 0.14$ (grey regime in Fig. 1), there is insufficient data to learn either set of weights –as signaled by vanishing overlaps $Q_{11} = Q_{22} = 0$–. As a consequence, the prediction error (black) stays sensibly constant, and does not decrease. In the regime $\alpha_1 < \alpha < \alpha_2 \approx 0.79$ (red regime in Fig. 1), the sample complexity is greater than the sharp threshold for the learnability of the second layer ($l = 2$), and the corresponding overlap becomes non-zero $Q_{22} \neq 0$, reflecting a partial reconstruction of $\boldsymbol{w}_2$. This translates into a sizable decrease in the prediction error throughout this phase. Finally, for sample complexities $\alpha > \alpha_2$ (blue regime in Fig. 1), the first layer weights are also learned ($Q_{11} \neq 0$), resulting in yet another decrease in prediction error. The different layers are thus learned sequentially, with a clear separation in sample complexity between the consecutive stages. The weak recovery thresholds $\alpha_1, \alpha_2$ delineating the different regimes can be computed using the results described in Section 2. Specifically, the initial recovery threshold $\alpha_1$ is the critical sample complexity above which a first subspace –in this case the span of the second layer weights– is recovered, as characterized in Theorem 2.3. $\alpha_2$ is the critical sample complexity required to recover the subspace spanned by the first layer weights, conditioned on having acquired a non-zero overlap $Q_{22} > 0$ with the second layer weights. We describe how to compute $\alpha_2$ in Appendix E.2. Finally, we chart the boundaries of the three regimes observed in Fig. 1 (left) in a phase diagram (Fig. 1 (right)), as a function of the skip connection $c$ and the sample complexity, evidencing how the phases subsist across a wide range of skip connection strengths $c$.

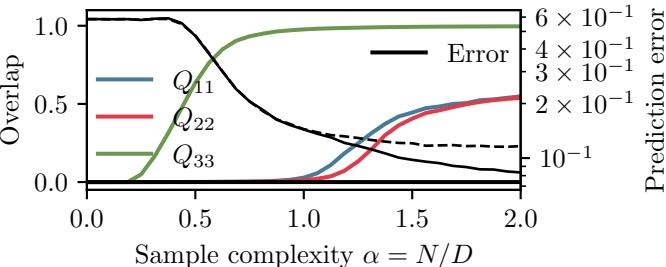 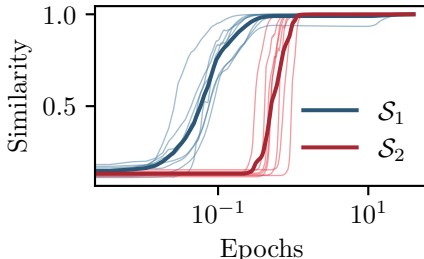

*Figure 3.* (**Left**) Diagonal elements of the overlap $Q$ (13) (red, blue and green) and prediction error (black) achieved by GAMP, as a function of the sample complexity $\alpha = {}^N/_D$, for a three-layer attention model $P_1 = P_2 = P_3 = 1, c = 1, M = 2$. Off-diagonal elements of $Q$ are zero, and thus not plotted. Dashed line: prediction error when the first and second layer weights are fixed to zero. (**Right**) Similarity $\mathcal{S}_l = {}_{\mathrm{tr}}\big((w_l^{t\top} w_l^t)(w_l^{*\top} w_{l*})\big)/\|w_l^t\|^2\|w_l^*\|^2$ between the weights $w_l^t$ at training step $t$ and the last-iterate weights $w_l^*$ as a function of the training time. A transformer with $L = 2$ attention layers and a fully connected readout is trained on the TREC classification task (Hovy et al., 2001; Li & Roth, 2002). Different colors indicate different attention layers, with the multiple curves representing distinct runs.

### 3.2. The role of depth

It is natural to wonder whether such layer-wise learning is a generic feature of deep attention networks or simply a phenomenon proper to the two-layer case. Fig. 3 (left) shows the diagonal elements of the overlap $Q$ for a depth $L = 3$ attention model with $P_1 = P_2 = P_3 = 1, M = 2, c = 1$. As in the $L = 2$ case (see Fig. 1), the last layer is learned first as the number of samples increases. On the other hand, the shallower layers $l = 1, 2$ seem to share the same weak recovery threshold, and are learned sensibly at the same rate. We conjecture this to be a general feature of deeper attention models.

Let us make an interesting observation in this direction, which we leave as inspiration for future work. The deep attention model (2) admits an elegant limiting form in the joint limit of large depth $L \to \infty$ and large skip connection $c = \beta L$, with $\beta$ finite, provided the activation function is simultaneously rescaled as $\sigma(z) = \check{\sigma}({}^z/_c)$:

$$y_{\mathrm{DA}}(x) = \check{\sigma}\bigg(\Big(1 + \tfrac{1}{\beta L}\textstyle\sum_{l=1}^{L-1}\sigma(Z_l^\top Z_l)\Big)^\top Z_L^\top Z_L\Big(1 + \tfrac{1}{\beta L}\textstyle\sum_{l=1}^{L-1}\sigma(Z_l^\top Z_l)\Big)\bigg),$$

where we remind that $Z_l = {}^{w_l x}/\sqrt{D}$ denotes the projection of the input $x$ on the $l-$th layer weights $w_l$. A detailed derivation of this limiting form is provided in Appendix A.4. This simple expansion shows that, in the large depth limit, the deep attention model can be viewed as a combination of a linear attention module $Z_L^\top Z_L$ parametrized by the last layer weights, and a "committee" of attention modules corresponding to the first $L - 1$ layers. Since these first layers are interchangeable we expect that also the overlaps should share the same invariance, making it possible to simplify the state evolution equations in this limit by assuming that also the overlaps need to have an interchangeable structure, in a similar spirit to (Aubin et al., 2018).

### 3.3. Dynamics of learning

The precedent subsection evidenced how the different layers warrant different sample complexities to be reconstructed and are learned sequentially by GAMP as more samples become available. A similar sequential learning phenomenon can in fact also be observed at a fixed sample complexity, *as a function of the number of iterations.* Fig. 2 (top) represents the evolution of the cosine similarity $\mathcal{C}_l = w_l^\star \hat{w}_l^\top / \|\hat{w}_l\|$ between the GAMP estimate $\hat{w}_l$ of the layer $l$ weights and the corresponding target weights $w_l^\star$, over the GAMP iterations. The second layer is learned within the first few iterations. In contrast, the first layer is at first not learned, and its reconstruction only commences after approximately 10 steps, when the second layer is almost perfectly reconstructed.

This layer-wise learning phenomenon over training time is not restricted to the GAMP algorithm. In Fig. 2 (bottom), we plot the performance of Stochastic Gradient Descent (SGD) in reconstructing the weights of the same two-layer architecture, from the same dataset $\mathcal{D} = \{x^\mu, y_{DA}^{w_1^\star, w_2^\star}(x^\mu)\}_{\mu=1}^N$, by implementing the descent steps

$$\hat{w}_l^{t+1} = -\eta \sum_{x \in B_t} \nabla_{\hat{w}_l}\|y_{DA}^{w_1^\star, w_2^\star}(x) - y_{DA}^{\hat{w}_1^t, \hat{w}_2^t}(x)\|^2 - \lambda\hat{w}_l^t,$$

with learning rate $\eta$ and weight decay $\lambda$. $B_t \subset \mathcal{D}$ denotes the batch used for the $t-$th step. Notice how we reuse the same batch for multiple gradient updates (Arnaboldi et al., 2024a). The simulations reveal that, similarly to GAMP, the learning of SGD proceeds in two sequential steps. While SGD starts learning the second layer weights after one epoch, the first layer is only reconstructed later, when the second layer is almost perfectly learned, with a clear separation in training time between the two stages. We believe that describing analytically the gradient descent dynamics of SMI models, and in particular of deep atten-

tions, can be an interesting research direction, which can be pursued leveraging the existing literature on the dynamics of multi-index models (Gerbelot et al., 2024; Celentano et al., 2021; Dandi et al., 2024; Mignacco et al., 2020). Similarly, a systematic study of layer-wise learning in more realistic transformers could offer valuable insights, especially though the lenses of mechanistic interpretability.

## 4. Discussion and conclusions

We have so far focused our investigations on the case of target functions of the form (2), evidencing how the weights of different layers of a two-layer attention architecture are learned sequentially over training time. In this last section, we briefly illustrate how similar layer-wise learning phenomena can also be observed for more practical transformer architectures, with real datasets.

We consider a classification problem on the Text REtrieval Conference (TREC) dataset (Hovy et al., 2001; Li & Roth, 2002), in which the network is tasked with labeling the object of a question. The data is pre-processed using the uncased base BERT model (Kenton & Toutanova, 2019) to obtain token embeddings that are subsequently fed into a simple transformer architecture with two consecutive attention layers, and a fully-connected readout layer. The network is trained on the cross-entropy loss using the `Pytorch` (Paszke et al., 2017) implementation of the AdamW (Loshchilov & Hutter, 2019) optimizer, over 40 epochs. Further details can be found in Appendix F. Fig. 3 shows the evolution of the similarity metric

$$\mathcal{S}_l = \frac{\mathrm{tr}\big((\boldsymbol{w}_l^{t\top}\boldsymbol{w}_l^t)(\boldsymbol{w}_l^{*\top}\boldsymbol{w}_{l*})\big)}{\|\boldsymbol{w}_l^t\|^2\|\boldsymbol{w}_l^*\|^2} \qquad (19)$$

between the weights $\boldsymbol{w}_l^t$ at training time $t$ and the last-iterate weights $\boldsymbol{w}_l^*$, for a given layer $l \in \{1, 2\}$, as a function of $t$. The curves reveal a marked sequential learning process. They are furthermore quite reminiscent of our findings for two-layer attention target functions, gathered in Fig. 2, which we discussed above. Remarkably, however, compared to Fig. 2, the order in which the layers are learned in the TREC task is reversed, echoing the findings of (Chen et al., 2023) that shallower layers tend to converge faster over training. We expect the order in which layers are learned to generically depend on the intricate interactions between the architecture, task and algorithm. Ascertaining precisely the factors determining the order in a given setting is a challenging task, which warrants careful mechanistic studies. We leave this question as an exciting future research direction.

To conclude, in the present manuscript, we consider the problem of learning a deep attention network, defined as the composition of multiple self-attention layers. We first establish that such architectures pertain to the class of sequence multi-index models, which generalize multi-index

models to sequential covariates. For this larger class of models, in the Bayes-optimal setting, we derive the statistically and computationally optimal estimation errors, and extract from those the weak recovery thresholds. In the case of a depth 2 attention model, our findings uncover how distinct layers are learned sequentially, with increasing number of samples, or over training time. We briefly discuss how a related layer-wise learning phenomenon can also be observed in practical settings.

Among the limitations of our work, the SMI functions do not naturally account for fully connected layers usually interlaid with attention layers. Another limitation is that we consider only learning from Gaussian i.i.d. inputs; adding input correlation is possible but left for future work.

### Acknowledgments

We thank Freya Behrens, Vittorio Erba, Mikhail Terekhov, and Fabrizio Boncoraglio for discussions. We acknowledge funding from the Swiss National Science Foundation grants SNFS SMArtNet (grant number 212049), and OperaGOST (grant number 200021_200390). HC acknowledges support from the Center of Mathematical Sciences and Applications (CMSA) of Harvard University.

## Impact statement

The goal of this work is to advance the field of theory of Machine Learning. There are multiple potential societal impacts which can follow, none of which we feel needs to be particularly highlighted.

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

## A. Generalization to untied and multi-head attention

The backbone of the analysis of deep attention models (2) reported in Section 3.1 of the main text is the mapping (7) to a SMI model (1). As we extensively discuss in the main text, this formal connection permits the transfer of analytical ideas from the study of multi-index models to that of deep attention architectures. Although we considered in the main text attention architectures with tied query and key weights for clarity and ease of exposition, the class of SMI functions (1) in fact encompasses a much broader array of deep attention architectures, with untied weights, and generic number of heads. In this Appendix, we detail the mapping of these more complex architectures to SMI functions. A direct consequence of this mapping is that the general results on the statistical (Theorem 2.1) and computational (Lemma 2.2 and Theorem 2.3) limits of learning in SMI models directly apply to such architectures. We discuss sequentially the case of deep attention functions with untied weights, before extending the mapping to multiple heads. Finally, we derive the infinite depth limit $L \to \infty$ discussed in the main text. In the following we will use bold characters for all the quantities that with $\mathcal{O}(D)$ elements, as we did in the main text.

### A.1. Untied weights

As such, our main theoretical results consequently hold for untied architectures as well. In this paragraph, we detail the mapping of a deep attention network with generically untied weights to a SMI. Consider the model

$$y_{\text{UDA}}(\boldsymbol{x}) = \sigma\left(\frac{\boldsymbol{x}_{L-1}^\top \boldsymbol{w}_L^{Q\top} \boldsymbol{w}_L^K \boldsymbol{x}_{L-1}}{D}\right), \qquad \forall l \in [\![1, L-1]\!], \ \boldsymbol{x}_l = \boldsymbol{x}_{l-1}\left[c\mathbb{1}_M + \sigma\left(\frac{\boldsymbol{x}_{l-1}^\top \boldsymbol{w}_l^{Q\top} \boldsymbol{w}_l^K \boldsymbol{x}_{l-1}}{D}\right)\right], \quad (20)$$

where $\boldsymbol{w}_l^Q, \boldsymbol{w}_l^K \in \mathbb{R}^{P_l \times D}$ are two sets of independently sampled standard Gaussian weights. As we did in the main text, we can write the untied deep attention model $y_{\text{UDA}}(\boldsymbol{x})$ as an SMI model by introducing two families of indices $\mathcal{K}_l, \mathcal{Q}_l$ for the keys and the queries respectively

$$\mathcal{K}_l = \frac{\boldsymbol{w}_l^K \boldsymbol{x}}{\sqrt{D}}, \qquad \mathcal{Q}_l = \frac{\boldsymbol{w}_l^Q \boldsymbol{x}}{\sqrt{D}}, \qquad (21)$$

with which we can define a new mixing functions $\{B_c^l(\mathcal{K}_1, ..., \mathcal{K}_l, \mathcal{Q}_1, ..., \mathcal{Q}_l)\}_{l=1}^L$, defined recursively as

$$B_c^l = (B_c^{l-1})^{(1-\delta_{l,L})}\left[(1 - \delta_{l,L})c\mathbb{1}_M + \sigma\left(B_c^{l-1\top}\mathcal{Q}_l^\top \mathcal{K}_l B_c^{l-1}\right)\right], \qquad (22)$$

with $B_c^0 = \mathbb{1}_M$, giving us

$$y_{\text{UDA}}(\boldsymbol{x}) = B_c^L(\mathcal{K}_1, ..., \mathcal{K}_L, \mathcal{Q}_1, ..., \mathcal{Q}_L) \qquad (23)$$

This corresponds to an SMI model (1). To see this, denote $P = P_1 + ... + P_L$ and introduce the concatenated weights $\boldsymbol{W}_{\text{UDA}} \in \mathbb{R}^{2P \times D}$, defined as the matrix obtained by vertically stacking $\boldsymbol{w}_1^K, ..., \boldsymbol{w}_L^K, \boldsymbol{w}_1^Q, ..., \boldsymbol{w}_L^Q$, in this order. Then define the link function $g_{\text{UDA}} : \mathbb{R}^{2P \times M} \to \mathbb{R}^{M \times M}$ as

$$g_{\text{UDA}}(X) = B_c^L(X_{[1:P_1]}, X_{[P_1+1:P_1+P_2]}, ..., X_{[P+1:P+P_1]}, ...), \qquad (24)$$

where $X_{[i:j]}$ denotes the $(j - 1 + 1) \times D$ submatrix of $X$ comprising the rows indexed between $i$ and $j$. Then, the deep attention architecture $y_{\text{UDA}}$ can be rewritten compactly as

$$y_{\text{UDA}}(\boldsymbol{x}) = g_{\text{UDA}}(\boldsymbol{W}_{\text{UDA}}\boldsymbol{x}). \qquad (25)$$

In other words, the untied deep attention is also a SMI. Consequently, all the technical results reported in the main text directly apply to this architecture.

### A.2. Sequence to Sequence multi-layer attention models

The SMI class of models defined by Eq. (1) further subsume sequence-to-sequence models with output in $\mathbb{R}^{D \times M}$ given by:

$$\mathbf{y} = \boldsymbol{x}_{l-1}\left[c\mathbb{1}_M + \sigma\left(\frac{\boldsymbol{x}_{l-1}^\top \boldsymbol{w}_l^\top \boldsymbol{w}_l \boldsymbol{x}_{l-1}}{D}\right)\right],$$

instead of the $\mathbb{R}^{M \times M}$ output in Eq. (2). Above, $\boldsymbol{x}_l$ are defined recursively as in Eq. (3) for $l \in [L]$.

The mapping of the above model to SMI follows by writing

$$\mathbf{y} = \boldsymbol{x} \bar{B}_c^L(Z_1, ..., Z_l), \tag{26}$$

where the functions $\bar{B}_c^l$ are the equivalent for sequence to sequence models of $B_c^l$ in (5)

$$\bar{B}_c^l(Z_1, ..., Z_l) = \bar{B}_c^{l-1}(Z_1, ..., Z_{l-1}) \Big[(1 - \delta_{l,L})c\mathbb{1}_M + \sigma\big(\bar{B}_c^{l-1}(Z_1, ..., Z_{l-1})^\top Z_l^\top Z_l \bar{B}_c^{l-1}(Z_1, ..., Z_{l-1})\big)\Big], \tag{27}$$

and $\bar{B}_c^1 = \mathbb{1}_M$. Intuitively $\bar{B}_c^L(Z_1, ..., Z_l) \in \mathbb{R}^{M \times M}$ denotes the components of the tokens (columns) in $\mathbf{y}$ along the basis of the input tokens in $\boldsymbol{x}$. Since we assume $D \gg P$, while $\mathbf{y}$ itself is high-dimensional, it can be constructed from $\boldsymbol{x}$ by specifying the $M \times M$ scalar entries of $\bar{B}_c^L(Z_1, ..., Z_l)$. Hence, the estimation of $\mathbf{y}$ given $\boldsymbol{x}$ is equivalent to the estimation of $\bar{B}_c^L(Z_1, ..., Z_l)$ given $\boldsymbol{x}$, which reduces to a sequence multi-index (SMI) model with outputs in $\mathbb{R}^{M \times M}$.

Note that for a single layer $L = 1$ this sequence-to-sequence variant of the model was analyzed in (Cui et al., 2024a).

## A.3. Multi-head attention

By the same token, multi-head architectures can also be mapped to a SMI, and thus fall within the scope of our theoretical results. Let us take the input data $\boldsymbol{x} \in \mathbb{R}^{DH \times M}$ sequence of length $M$ embedded in dimension $DH$. At each layer $l$, the embeddings are split into $H$ vectors of dimension $D$, which are then projected to dimension $P_l$. We can model this by first splitting $\boldsymbol{x}$ into a list of $H$ elements $\{\boldsymbol{x}^h \in \mathbb{R}^{D \times M}\}_{h=1}^H$ and then defining $H \times H$ sequence indices for each layer both for the keys and the queries

$$\mathcal{K}_l^{h_1 h_2} = \frac{{}^K\boldsymbol{w}_l^{h_1 h_2} \boldsymbol{x}^{h_2}}{\sqrt{D}}, \qquad \mathcal{Q}_l^{h_1 h_2} = \frac{{}^Q\boldsymbol{w}_l^{h_1 h_2} \boldsymbol{x}^{h_2}}{\sqrt{D}} \tag{28}$$

Each attention head is defined as

$$\sigma\left(\sum_{h_1, h_2}^H \frac{\boldsymbol{x}_{l-1}^{h_1 \top} {}^Q\boldsymbol{w}_l^{hh_1 \top} {}^K\boldsymbol{w}_l^{hh_2} \boldsymbol{x}_{l-1}^{h_2}}{HD}\right) = \sigma\left(\frac{1}{H}\sum_{h_1, h_2}^H (\mathcal{Q}_l^{hh_1})^\top \mathcal{K}_l^{hh_2}\right). \tag{29}$$

The deep multi-head architecture is then defined as

$$\forall l \in [\![1, L]\!], \ \boldsymbol{x}_l^h = \boldsymbol{x}_{l-1}^h \left[(1 - \delta_{l,L})c\mathbb{1}_M + \sigma\left(\frac{1}{H}\sum_{h_1, h_2}^H (\mathcal{Q}_l^{hh_1})^\top \mathcal{K}_l^{hh_2}\right)\right] \tag{30}$$

with

$$y_{\text{MHA}}(\boldsymbol{x}) = \sum_h \boldsymbol{v}_h^\top x_L^h, \tag{31}$$

where $\boldsymbol{v}_h \in \mathbb{R}^{P_r \times D}$ are trainable readout weights. As before, we introduce a set of functions $B_c^l : (\mathbb{R}^{H \times H \times P_1 \times M})^{\otimes 2} \times ... \times (\mathbb{R}^{H \times H \times P_l \times M})^{\otimes 2} \to \mathbb{R}^{H \times M \times M}$

$$[B_c^l]_h = [B_c^{l-1}]_h^{(1-\delta_{l,L})} \left[(1 - \delta_{l,L})c\mathbb{1}_M + \sigma\left(\sum_{h_1, h_2}^H [B_c^{l-1}]_{h_1}^\top (\mathcal{Q}_l^{hh_1})^\top \mathcal{K}_l^{hh_2} [B_c^{l-1}]_{h_2}\right)\right], \tag{32}$$

with $[B_c^0]_h = \mathbb{1}_M$. Then

$$y_{\text{MHA}}(\boldsymbol{x}) = \sum_h \mathcal{V}_h [B_c^L(\{\{\mathcal{K}_l^{h1}, ..., \mathcal{K}_l^{hH}\}_{h=1}^H\}_{l=1}^L, \{\{\mathcal{Q}_l^{h1}, ..., \mathcal{Q}_l^{hH}\}_{h=1}^H\}_{l=1}^L)]_h, \tag{33}$$

defining $\mathcal{V}_h \equiv \boldsymbol{v}_h^\top \boldsymbol{x}_h$.

To expound the connection of this deep, multi-head model to SMI models, first denote $P = (P_1 + ... + P_L)H + P_r$, and define the total weights $\boldsymbol{W}_{\text{MHA}} \in \mathbb{R}^{P \times DH}$, defined as

$$\boldsymbol{W}_{\text{MHA}}^\top = \big({}^K\boldsymbol{W}_1^1 ... {}^K\boldsymbol{W}_1^H ... {}^K\boldsymbol{W}_L^1 ... {}^K\boldsymbol{W}_L^H \ {}^Q\boldsymbol{W}_1^1 ... {}^Q\boldsymbol{W}_1^H ... {}^Q\boldsymbol{W}_L^1 ... {}^Q\boldsymbol{W}_L^H \ \boldsymbol{V}\big). \tag{34}$$

with

$$
{}^{K}\boldsymbol{W}_l^h = \begin{pmatrix} {}^{K}\boldsymbol{w}_1^{h1} & & 0 \\ & \ddots & \\ 0 & & {}^{K}\boldsymbol{w}_1^{hH} \end{pmatrix}^{\top} \quad {}^{Q}\boldsymbol{W}_l^h = \begin{pmatrix} {}^{Q}\boldsymbol{w}_1^{h1} & & 0 \\ & \ddots & \\ 0 & & {}^{Q}\boldsymbol{w}_1^{hH} \end{pmatrix}^{\top} \quad \boldsymbol{V} = \begin{pmatrix} \boldsymbol{v}_1 & & 0 \\ & \ddots & \\ 0 & & \boldsymbol{v}_H \end{pmatrix}^{\top} \tag{35}
$$

Finally, we construct the link function $g_{\text{MHA}} : \mathbb{R}^{P \times M} \to \mathbb{R}^{P_r \times M}$. For any argument $X \in \mathbb{R}^{P \times M}$, we view $X$ as a vertical block vector with $2HL + H$ blocks $\{\{{}^{K}X_l^h \in \mathbb{R}^{HP_l \times M}\}_{h=1}^{H}\}_{l=1}^{L}, \{\{{}^{Q}X_l^h \in \mathbb{R}^{HP_l \times M}\}_{h=1}^{H}\}_{l=1}^{L}, \{{}^{V}X \in \mathbb{R}^{HP_r \times M}\}$, in this order. Each of these blocks is further viewed as a block matrix of $H$ vertically stacked blocks, which we index with the subscript $h'$. Then we define

$$
g_{\text{MHA}}(X) = \sum_h {}^{V}X_h [B_c^L(\{\{({}^{K}X_l^h)_1, ..., ({}^{K}X_l^h)_H\}_{h=1}^{H}\}_{l=1}^{L}, \{\{({}^{Q}X_l^h)_1, ..., ({}^{Q}X_l^h)_H\}_{h=1}^{H}\}_{l=1}^{L})]_h. \tag{36}
$$

With those definitions, it follows that the multi-head deep attention model can be rewritten as

$$
y_{\text{MHA}}(\boldsymbol{x}) = g_{\text{MHA}}(\boldsymbol{W}_{\text{MHA}}\boldsymbol{x}), \tag{37}
$$

i.e. this model too pertains to the class of SMI models. All the technical results established for SMI models in the main text thus directly transfer to multi-head architectures.

This Appendix showcases how fairly intricate deep attention architectures, with generically untied weights and multiple heads, fall into the class of SMI models – which thus proves a truly rich class of models. The study of the phenomenology of these more complex models and the implication of our theoretical characterization, therefore, are left for future works.

### A.4. Large depth limit of tied, single-head attention

In this subsection we study the limit of strong skip connection in the model we are considering, showing how the formalism simplifies in this limit. We start by redefining the multi-layer attention by rescaling each layer

$$
\forall l \in [\![1, L-1]\!], \qquad \boldsymbol{x}_l = \boldsymbol{x}_{l-1} \left[ \mathbb{1}_M + \frac{1}{c}\sigma\left( \frac{\boldsymbol{x}_{l-1}^{\top} \boldsymbol{w}_l \boldsymbol{w}_l^{\top} \boldsymbol{x}_{l-1}}{D} \right) \right]. \tag{38}
$$

With these new definitions, we are interested in the large depth and skip connection limit $c, L \to \infty$, while keeping the two terms proportional by imposing $c = \beta L$. We start by writing the functions $B_c^l$ in this new definition

$$
B_c^l(Z_1, ..., Z_l) = B_c^{l-1}(Z_1, ..., Z_{l-1}) \left[ \mathbb{1}_M + \frac{1}{c}\sigma\left( B_c^{l-1}(Z_1, ..., Z_{l-1})^{\top} Z_l^{\top} Z_l B_c^{l-1}(Z_1, ..., Z_{l-1}) \right) \right], \tag{39}
$$

with $B_c^0 = \mathbb{1}_M$. We would like to expand the functions above for large $c$, so we define the functions $A_c^l$ such that

$$
B_c^l = \mathbb{1}_M + \frac{A_c^l}{c} + \mathcal{O}\left(\frac{1}{c}\right). \tag{40}
$$

For $l = 0$ we have $A_c^0 = 0$. It is possible to determine the other values of $A_c^l$ by expanding (39)

$$
\begin{aligned}
B_c^l &= B_c^{l-1}\left[ \mathbb{1}_M + \frac{1}{c}\sigma\left( (B_c^{l-1})^{\top} Z_l^{\top} Z_l B_c^{l-1} \right) \right] \\
&= \left( \mathbb{1}_M + \frac{1}{c}A_c^{l-1} \right) \left[ \mathbb{1}_M + \frac{1}{c}\sigma\left( \left( \mathbb{1}_M + \frac{1}{c}A_c^{l-1} \right)^{\top} Z_l^{\top} Z_l \left( \mathbb{1}_M + \frac{1}{c}A_c^{l-1} \right) \right) \right] + \mathcal{O}\left(\frac{1}{c}\right) \\
&= \mathbb{1}_M + \frac{1}{c}A_c^{l-1} + \frac{1}{c}\sigma\left( \left( \mathbb{1}_M + \frac{1}{c}A_c^{l-1} \right)^{\top} Z_l^{\top} Z_l \left( \mathbb{1}_M + \frac{1}{c}A_c^{l-1} \right) \right) + \mathcal{O}\left(\frac{1}{c}\right) \\
&= \mathbb{1}_M + \frac{1}{c}A_c^{l-1} + \frac{1}{c}\sigma\left( Z_l^{\top} Z_l \right) + \mathcal{O}\left(\frac{1}{c}\right),
\end{aligned} \tag{41}
$$

from which we can extract

$$A_c^l = A_c^{l-1} + \sigma\left(Z_l^\top Z_l\right) . \tag{42}$$

We can thus provide a closed form expression for $A_c^l$

$$\forall l \in [\![1, L-1]\!], \qquad A_c^l = \sum_{l'=1}^{l} \sigma\left(Z_{l'}^\top Z_{l'}\right) . \tag{43}$$

Finally, we have the result

$$B_c^L(Z_1, ..., Z_L) = \sigma\left(\left(\left(1 + \frac{1}{\beta L}\sum_{l=1}^{L-1}\sigma\left(Z_l^\top Z_l\right)\right)^\top Z_L^\top Z_L \left(1 + \frac{1}{\beta L}\sum_{l=1}^{L-1}\sigma\left(Z_l^\top Z_l\right)\right)\right)\right) \tag{44}$$

This is a very special function. In order to highlight its properties, we will study a generic functions $g : \mathbb{R}^{M \times M} \times \mathbb{R}^{M \times M} \to \mathbb{R}^{M \times M}$

$$g(J, H) = \sigma\left(\left(1 + \frac{H}{\beta}\right)^\top J \left(1 + \frac{H}{\beta}\right)\right) , \tag{45}$$

with the identification

$$B_c^L(Z_1, ..., Z_L) = g\left(Z_L^\top Z_L, \frac{1}{L}\sum_{l=1}^{L-1}\sigma\left(Z_l^\top Z_l\right)\right) \tag{46}$$

## B. Proof of Theorem 2.1

In this Appendix, we detail the equivalence, discussed in subsection 2.1 of the main text, between a SMI function (1) and a classical multi-index model acting on flattened input and with shared weights. This connection is instrumental in the application of the results of (Gerbelot & Berthier, 2023; Aubin et al., 2018; Troiani et al., 2025) for multi-index models to SMI models. Leveraging this equivalence, we then provide a proof of Theorem 2.1.

### B.1. Mapping SMI models to block-structured multi-index models

Consider the setup of sequence multi-index models defined by Eq. (1) and suppose that the link function $g : \mathbb{R}^{P \times M} \to \mathbb{R}^K$ outputs $K$-dimensional vectors $y_{\boldsymbol{W}^\star}^{\mathrm{SMI}}(\boldsymbol{x})$.

Given a sequence $\boldsymbol{x} \in \mathbb{R}^{D \times M}$, we denote by $\boldsymbol{x}_e \in \mathbb{R}^{DM}$ the corresponding "flattened" vector defined by:

$$\boldsymbol{x}_e^\top \coloneqq [\boldsymbol{x}_1^\top, \cdots, \boldsymbol{x}_M^\top], \tag{47}$$

where $\boldsymbol{x}_1, \cdots, \boldsymbol{x}_M$ denote the columns of $\boldsymbol{x}$

Under the above flattening, the pre-activations $\boldsymbol{W}\boldsymbol{x} \in \mathbb{R}^P$ can be re-expressed through projections of the flattened inputs $x_e$ onto a block-structured weight matrix given by:

$$\boldsymbol{W}_e^\star \coloneqq \begin{bmatrix} \boldsymbol{W}^\star, & 0, & \cdots, & 0 \\ 0, & \boldsymbol{W}^\star, & \cdots, & 0 \\ \vdots & \vdots & \vdots & \vdots \end{bmatrix} \in \mathbb{R}^{PM \times DM}. \tag{48}$$

Equivalently, $\boldsymbol{W}_e^\star$ can be represented as the Kronecker product $\boldsymbol{W}^\star \otimes \mathbf{I}_M$

The above representation leads to the following correspondence:

$$(\boldsymbol{W}_e^\star \boldsymbol{x}_e) = \begin{bmatrix} (\boldsymbol{W}^\star \boldsymbol{x}_1) \\ \cdots \\ (\boldsymbol{W}^\star \boldsymbol{x}_M) \end{bmatrix}, \tag{49}$$

Therefore, the $PM$-dimensional vector $(\boldsymbol{W}_e^\star \boldsymbol{x}_e)$ is obtained by stacking the projections of different columns of $\boldsymbol{x}$ onto $\boldsymbol{W}^\star$.

With the above definition, $y_{\boldsymbol{W}^\star}^{\mathrm{SMI}}$ can be equivalently expressed as:

$$y_{\boldsymbol{W}^\star}^{\mathrm{SMI}} = g\left(\frac{\boldsymbol{W}^\star \boldsymbol{x}}{\sqrt{D}}\right) = g_e\left(\frac{\boldsymbol{W}_e^\star \boldsymbol{x}_e}{\sqrt{D}}\right), \tag{50}$$

where $g_e : \mathbb{R}^{PM} \to \mathbb{R}^K$ is obtained by lifting $g$ through the mapping defined by Eq. (49) to act on inputs in $\mathbb{R}^{PM}$.

## B.2. Proof of Theorem 2.1

We make the following assumption on $g$ derived from (Aubin et al., 2020; Barbier et al., 2019):

**Assumption B.1.** The non-linearity $g$ in SMI (model 1) lies in $\mathcal{C}^2$ and is entry-wise bounded.

For technical reasons, we further assume that a Gaussian noise matrix with i.i.d $\mathcal{N}(0, \Delta)$ entries with arbitrarily small variance $\Delta$ is added to the SMI output (1) and the corresponding channel (9):

**Assumption B.2.** We consider the limit $\Delta \to 0$ under the modified activation $\tilde{g}(\cdot) = g(\cdot) + \sqrt{\Delta}\xi$,

where $\xi \sim \mathcal{N}(0, \mathbb{I}_K)$ denote noise independent across samples.

The above assumptions are required due to the limitations of the adaptive interpolation framework in (Aubin et al., 2020; Barbier et al., 2019) and we believe they can be relaxed through a more careful control of the error terms.

We establish Theorem 2.1 by combining the above mapping defined by Eq. (50) with the adaptive interpolation scheme in (Aubin et al., 2020; Barbier et al., 2019). The central idea behind the adaptive interpolation scheme is to interpolate between the high-dimensional Hamiltonian corresponding to the posterior measure (Eq. 8) and an effective "factorized" Hamiltonian constructed through the low-dimensional output channel defined by Eq. (9) and an effective input channel on $\mathbb{R}^P$. Analogous to Gaussian-comparison inequalities, one then utilizes Stein's Lemma to relate the time-derivative of the free-energy over the interpolation path to the overlaps corresponding to the Hamiltonian at time $t$. Using the concentration of overlaps (justified through the Nishimori identity and the inclusion of vanishing side-information), one then obtains a family of interpolation paths satisfying systems of ODEs such that the high-dimensional remainder terms during the interpolation vanish. Selecting particular choices for such interpolation paths then yields matching lower and upper bounds for the free-energy.

The mapping of SMI models to block-structured multi-index models on flattened inputs given in Section B allows us to prove the asymptotic expression for the free-energy through a reduction to the steps utilized in the proof of Theorem 3.1 in (Aubin et al., 2020) and Theorems 1,2 in (Barbier et al., 2019). For brevity, we only describe the central differing step, related to the derivation of the time-derivative of the free-entropy described above.

Following (Aubin et al., 2020), we parameterize the interpolation paths through two functions $r(t) : [0,1] \to \mathcal{S}_P^+$ and $q(t) : [0,1] \to \mathcal{S}_P^+$ and scalar parameters $\epsilon_1, \epsilon_2 \in \mathbb{R}^+$. Let $R_1(t), R_2(t)$ denote the solutions to the following ODEs:

$$R_1(t) \coloneqq \epsilon_1 + \int_0^t r(t)\,\mathrm{d}s, \qquad R_2(t) \coloneqq \epsilon_1 + \int_0^t q(t)\,\mathrm{d}s \tag{51}$$

Define $S_{t,\mu} \in \mathbb{R}^{P \times M}$ for $\mu \in [N]$ as the following "interpolating" inputs to the effective output channel (Eq. (9)):

$$S_{t,\mu} \coloneqq \sqrt{1-t}\frac{\boldsymbol{W}\boldsymbol{x}}{\sqrt{D}} + \sqrt{R_2(t)}\omega_\mu + \sqrt{t\mathbb{1} - R_2(t)}U_\mu^\star, \tag{52}$$

where $\omega_\mu, U_\mu^\star \in \mathbb{R}^{P \times M}$ have entries $\overset{\text{i.i.d.}}{\sim} \mathcal{N}(0,1)$, with $\omega_\mu$ playing the role of observed side-information while $\boldsymbol{W}, U_\mu^\star$ denote unkown parameters.

We further define the corresponding interpolating variables at the input channel for $i \in [D]$:

$$Y_{t,i}' = \sqrt{R_1(t)}\boldsymbol{W}_i + Z_i', \tag{53}$$

where $Y_{t,i}', X_i^\star \in \mathbb{R}^P$. We thus obtain two sets of "interpolating" parameters, observations:

$$Y_{t,\mu} = g(S_{t,\mu}), \tag{54}$$

for $\mu \in [N]$ and $Y'_{t,i}$ for $i \in [D]$. These define an "interpolating posterior":

$$P(\boldsymbol{W}, U^\star | \{Y'_{t,i}\}_{i \in [D]}, \{Y_{t,\mu}, \boldsymbol{x}_\mu, \omega_\mu\}_{\mu \in [N]}) = \frac{1}{Z_{n,\epsilon}(t)} e^{-\frac{1}{2} \mathrm{Tr}[\boldsymbol{W}\boldsymbol{W}^\top]} \prod_{d=1}^D e^{-\frac{1}{2} \|Y'_{t,i} - \sqrt{R_1(t)}\boldsymbol{W}_i\|^2} \prod_{\mu=1}^N P_Y(Y_{t,\mu}|S_{t,\mu}). \quad (55)$$

, (Eq. 17 in (Aubin et al., 2018)) and the associated free-entropy:

$$f_{n,\epsilon}(t) = \frac{1}{n} \mathbb{E}[\log Z_{n,\epsilon}(t)], \quad (56)$$

which reduces to the original model (Equation 8) at $t, \epsilon = 0$.

The only difference from the setup in (Aubin et al., 2020) lies in that $S_{t,\mu}$ are matrices, instead of $P$-dimensional vectors. However, leveraging the flattening defined by Eq. (49), we may express $S_{t,\mu}$ as:

$$S_{t,\mu,e} := \sqrt{1-t}\frac{\boldsymbol{W}^\star_e \boldsymbol{x}_e}{\sqrt{D}} + \sqrt{R_2(t)}_e \omega_{\mu,e} + \sqrt{t\mathbb{1} - R_2(t)}_e U^\star_{\mu,e}, \quad (57)$$

Under the above flattening, equation 44 in (Aubin et al., 2018) (arising from applying Gaussian integration by parts to the interpolating Hamiltonian w.r.t variables $\boldsymbol{x}_e$ and $U^\star_{\mu,e}$) are modified with the replacement:

$$\boldsymbol{W}^\star \boldsymbol{W}^\top \to \boldsymbol{W}^\star_e \boldsymbol{W}^\top_e. \quad (58)$$

The mapping is then completed by noting that the flattening operator commutes under multiplication (as one expects from the properties of Kronecker/tensor products). Therefore:

$$\boldsymbol{W}^\star_e \boldsymbol{W}^\top_e = (\boldsymbol{W}^\star \boldsymbol{W}^\top)_e. \quad (59)$$

Under the above substitution, the limiting free-entropy is obtaining by following the remainder of the proof of Theorem 3.1 in (Aubin et al., 2018), resulting in the variational problem given by Eq. (10). Subsequently, the resulting prediction error is obtained through steps identical to the proof of Theorem 2 in (Barbier et al., 2019).

## C. Proofs of Lemma 2.2 and Theorem 2.3

In this section, we prove the results presented in Section 2.3, namely the tight asymptotic characterization of AMP (Algorithm 1) in terms of its finite-dimensional state evolution equations (Lemma 2.2), and the characterization of weak recovery thresholds (Theorem 2.3). We start by establishing Lemma 2.2 through a mapping to general results for GAMP algorithms with non-separable non-linearities. We remark that Lemma 2.2 and Theorem 2.3 themselves do not establish the optimality of Algorithm 1 or even prescribe its particular form – the latter will be derived in Appendix D. However, by comparing the resulting expressions for state-evolution (14) to the extremization criteria in Theorem 2.1, we observe that GAMP asymptotically achieves the optimal error.

Throughout, we make the following assumptions on the non-linearity $g$:

**Assumption C.1.** The denoiser $g_{out}$ defined by 12 lies in $\mathcal{C}^2$.

It is easy to check that the above assumptions holds for any $g \in \mathcal{C}^2$ in the presence of vanishing noise.

### C.1. Proof of Lemma 2.2

Analogous to the proof of Theorem 2.1, the proof of Lemma 2.2 relies on the mapping of the sequence models to multi-index models on flattened inputs, as described in Appendix B. Under this mapping, Algorithm 1 can be expressed in the following form:

$$\boldsymbol{\Omega}^t = \boldsymbol{x}_e f_t(\mathbf{B}^t) - g_{t-1}(\boldsymbol{\Omega}^{t-1}, \mathbf{y})\boldsymbol{V}_t \quad (60)$$

$$\mathbf{B}^{t+1} = \boldsymbol{x}_e^T g_t(\boldsymbol{\Omega}^t, \mathbf{y}) + f_t(\mathbf{B}^t)\boldsymbol{A}_t, \quad (61)$$

where $\mathbf{\Omega}^t \in \mathbb{R}^{N \times PM}$ contains rows corresponding to the flattened version of $\omega_e^t$ in Algorithm 1 as per Eq. 49 and $\mathbf{B^t} \in \mathbb{R}^{\mathbf{DM} \times \mathbf{PM}}$ contains columns given by the columns of $\hat{W}_e^t$ in Algorithm 1, flattened as per Eq. 48. $V_t, A_t$ then denote the corresponding "flattened" Onsager terms. Finally, the non-linearities $g_t$ and $f_t$ denote the flattened versions of $\partial_\omega g_{out}$ and $(\mathbb{1}_P + A^t)^{-1}$ respectively:

$$g_t(\mathbf{\Omega}^t, \mathbf{y})_\mu = \partial_\omega g_{out}(\omega_\mu^t, y_\mu)_e \in \mathbb{PM}, \tag{62}$$

applied row-wise and:

$$f_t(\mathbf{B^t}) = (\mathbb{1}_{\mathbf{P}} + \mathbf{A^t})_{\mathbf{e}}^{-1} \mathbf{B^t}, \tag{63}$$

where the notation $_e$ represents the flattening and tensorization defined in Equations 47, 48. We note that the block-structured multiplication in $f_t$, which maintains the same estimate $\hat{W}^t$ for different rows of $\hat{W}_e^t$ is non-separable across the $DM$ rows of $\mathbf{B^t}$. Such non-seperable non-linearities are however, permitted within the existing analysis in (Gerbelot & Berthier, 2023; Berthier et al., 2020) since it leads to well-defined convergence of empirical averages (Assumptions A5-A7 in Theorem 1 of (Gerbelot & Berthier, 2023)). Furthermore, using the commutativity of the tensorization and matrix-multiplication in (59), the resulting state-evolution equation (Eq. (14)) admits the same form as for the regular multi-index models in (Troiani et al., 2025).

Therefore, applying Theorem 1 in (Gerbelot & Berthier, 2023), we obtain that for any number of finite time step $t \in \mathbb{N}$:

$$\frac{\hat{W}^t(W^*)^\top}{D} \xrightarrow[P]{d \to \infty} Q^t, \tag{64}$$

where $Q^t$ satisfies Eq. 14.

## C.2. Definition of the Weak-Recovery Thresholds and the Proof of Theorem 2.3

As explained in the discussion preceding Theorem 2.3, Lemma 2.2 allows us to characterize weak-recovery under vanishing side-information through a stability analysis of the state-evolution. We begin by providing the precise definitions for weak-recovery thresholds based on (Troiani et al., 2025):

**Definition C.2.** Consider Algorithm 1 in the presence of side information of the form $\lambda W^\star + \sqrt{1 - \lambda}\xi$, where $\xi$ contains independent entries $\xi_{ij} \sim \mathcal{N}(0, 1)$. We say that $\alpha_c > 0$ is a *weak-recovery threshold* for Algorithm 1 if the following hold with high probability as $D \to \infty$:

1. For any $\alpha > \alpha_c$, there exists $\delta > 0$ such that for any (arbitrarily small) $\lambda > 0$, $\exists t = \mathcal{O}(\log \frac{1}{\lambda})$ such that $\left\| \frac{\hat{W}^t(W^\star)^\top}{D} \right\| \geq \delta$.

2. For $\alpha < \alpha_c$, $\left\| \frac{\hat{W}^t(W^\star)^\top}{D} \right\| \leq c\sqrt{\lambda}$ for some constant $c > 0$ and any $t \in \mathbb{N}$.

Analogously, we define thresholds characterizing the weak-recovery of given a subspace $U \subseteq \mathbb{R}^p$:

**Definition C.3.** For a subspace $U \subseteq \mathbb{R}^p$, we say that $\alpha_U > 0$ is the *subspace weak-recovery threshold* for $U$ if:

1. For any $\alpha > \alpha_U$, there exists $\delta > 0$ such that for any (arbitrarily small) $\lambda > 0$, $\exists t = \mathcal{O}(\log \frac{1}{\lambda})$ such that $\frac{\hat{W}^t W^\star}{D} \succ \delta P_U$.

2. For $\alpha < \alpha_U$, $\exists v \in U$ with $\|v\| = 1$ such that $\left\| \frac{\hat{W}^t v v^\top (W^\star)^\top}{D} \right\| \leq c\sqrt{\lambda}$ for some constant $c > 0$ for any $t \in \mathbb{N}$.

Under the above definitions and Lemma 2.2, the proof of Theorem 2.3 follows (Troiani et al., 2025). Since the form of state-evolution in Equation 18 exactly matches the form in (Troiani et al., 2025), we only outline the main steps.

The inclusion of side-information $\lambda W^\star + \sqrt{1 - \lambda}\xi$, which amounts to a modification of the prior over $W$, changes the state evolution dynamics in Lemma 2.2 to Equation 18. Under the assumption $g_{\text{out}} \in \mathcal{C}^2$ the following linearization holds for the state-evolution dynamics defined by Equation 18:

$$Q^{t+1} \approx \alpha \mathcal{F}(\delta Q^t) + \sqrt{\lambda}\mathbf{I}_d + \mathcal{O}(\alpha \|\delta Q^t\|_F^2) + \mathcal{O}(\lambda), \tag{65}$$

The term $\sqrt{\lambda}\mathbf{I}_d$ arising from side-information implies that at $t = 1$, the overlaps satisfy for small enough $\lambda$:

$$\mathrm{tr}(Q^1, Q^\star) > \lambda > 0, \tag{66}$$

with $Q^1 \in \mathbb{S}_P^+$. The generalized-Perron Frobenius theorem then implies that $\mathcal{F}$ admits a leading eigenvector in $\mathbb{S}_P^+$. Therefore, whenever $\alpha > \alpha_c$, we obtain a recursion of the form:

$$\mathrm{tr}(Q^{t+1}, Q^\star) \geq (1 + \kappa)\,\mathrm{tr}(Q^t, Q^\star), \tag{67}$$

for some $\kappa > 0$, leading to the positive part of Theorem 2.3 for $\alpha > \alpha_c$ through Theorem 4.2 in (Troiani et al., 2025). The negative part follows from an analogous reverse inequality as detailed in the proof of Theorem 4.2 in (Troiani et al., 2025). Similarly, the proof of Proposition 5.1 in (Troiani et al., 2025) based on the monotonocity of the GAMP dynamics, implies the existence and consistency with Definition C.3 of the subspace weak-recovery threshold defined by Equation 17.

## D. Derivation of GAMP from Belief Propagation

This Appendix details the derivation of the precise form of the GAMP algorithm (Algorithm 1) associated with the Bayesian inference problem (8), alongside the expressions of the state-evolution equations (14) that asymptotically describe its dynamics. A rigorous proof of the state evolution is presented in Appendix C.

The departure point of this derivation are the Belief Propagation iterations (Pearl, 1988) associated with (8), then simplifying them to offer a linear time algorithm that can be practically implemented with ease (Mezard & Montanari, 2009; Wainwright & Jordan, 2008; Lauditi et al., 2024). Contextually, we will derive the State Evolution iteration (14) and show that it describes the performance of GAMP at each iteration. Note that the derivation of GAMP and its state evolution for SMI models was previously reported in (Cui, 2025) for empirical risk minimization problems. We complement this analysis with the case of Bayesian inference considered in the present work. While this derivation is heuristic in nature, we feel it provides an insightful view of how to propose algorithms for Bayesian estimation.

We start by defining a factor graph, which is a bipartite graph with two families of nodes: $D$ variable notes, each encoding a weight vector $\boldsymbol{W}_i \in \mathbb{R}^P$ and $D + N$ factor nodes that connect them. $D$ of the factor nodes are each linked to one variable node and encode the Gaussian prior, the remaining $N$ are each connected to all the weights and represent the constraint that properly tuned weights should map the input data $\boldsymbol{x}^\mu \in \mathbb{R}^{D \times M}$ to the labels $y^\mu \in \mathbb{R}^{M'}$ based on the examples in the dataset through a sequence multi-index model linked by $g : \mathbb{R}^{P \times M} \to \mathbb{R}^{M'}$. We give a pictorial representation in Fig. 4. We then introduce two families of messages $\hat{m}_{\mu \to i}, m_{i \to \mu}$, which are the marginal probabilities of $\boldsymbol{W}_i$ if we remove the edges $i \to \mu$ or $\mu \to i$. In the following it will be convenient to call $\boldsymbol{W} \in \mathbb{R}^{P \times D}$ the stack of vectors $\boldsymbol{W}_i$

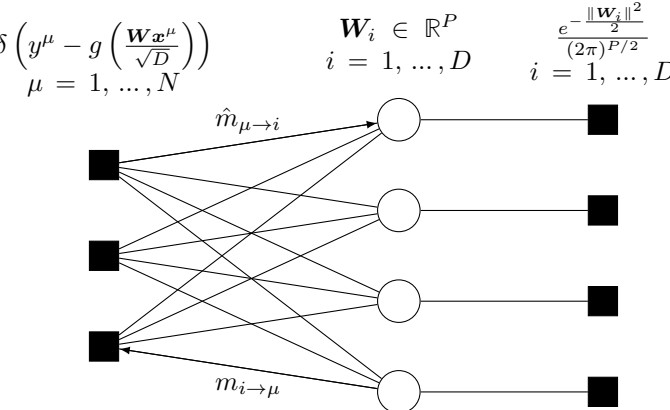

Figure 4. Factor graph representation of a sequence multi-index model with 4 variable nodes (circles) and $3 + 4$ factor nodes (squares). Picture inspired by (Aubin et al., 2018)

It is possible to define an iterative algorithm on the messages, commonly called the Belief Propagation (BP) equations.

$$m_{i\to\mu}^{t+1}(\boldsymbol{W}_i) = \frac{1}{\mathcal{Z}_{i\to\mu}} \frac{e^{-\frac{\|\boldsymbol{w}_i\|^2}{2}}}{(2\pi)^{P/2}} \prod_{\nu\neq\mu}^{N} \hat{m}_{\nu\to i}^t(\boldsymbol{W}_i)\,,$$

$$\hat{m}_{\mu\to i}^t(\boldsymbol{W}_i) = \frac{1}{\mathcal{Z}_{\mu\to i}} \int \prod_{j\neq i}^{D} \mathrm{d}\boldsymbol{W}_j \delta\left(y^\mu - g\left(\sum_{j=1}^{D} \frac{\boldsymbol{W}_j \boldsymbol{x}_j^\mu}{\sqrt{D}}\right)\right) m_{j\to\mu}^t(\boldsymbol{W}_j)\,.$$

(68)

It is impossible to iterate the BP equations for this problem in practice, but that's not an issue: to estimate the weights $w$ we only need the first few moments of the messages. This observation will allow us to obtain an efficient algorithm called Relaxed BP. It is convenient to introduce the sequence indices $Z^\mu \in \mathbb{R}^{P\times M}$ using a Fourier transform.

$$\delta\left(y^\mu - g\left(\sum_{j=1}^{D} \frac{\boldsymbol{W}_j \boldsymbol{x}_j^\mu}{\sqrt{D}}\right)\right) =$$

$$\frac{1}{(2\pi)^{PM}} \int_{\mathbb{R}^{P\times M}} \mathrm{d}Z^\mu \mathrm{d}\hat{Z}^\mu \exp\left\{-i\sum_{m=1}^{M}(\hat{Z}_m^\mu)^\top\left(\sum_{j=1}^{D}\frac{\boldsymbol{W}_j \boldsymbol{x}_{jm}^\mu}{\sqrt{D}}\right) + i\sum_{m=1}^{M} Z_m^{\mu\top}\hat{Z}_m^\mu\right\}\delta\left(y^\mu - g(Z^\mu)\right)\,.$$

(69)

This representation can be put in the equation (68) for $\hat{m}_{\mu\to i}^t(\boldsymbol{W}_i)$, taking care of separating the contribution of $\boldsymbol{W}_i$

$$\hat{m}_{\mu\to i}^t(\boldsymbol{W}_i) = \frac{1}{(2\pi)^{PL}\mathcal{Z}_{\mu\to i}} \int_{\mathbb{R}^{P\times M}} \mathrm{d}Z^\mu \mathrm{d}\hat{Z}^\mu \delta\left(y^\mu - g(Z^\mu)\right) \exp\left\{-i\sum_{m=1}^{M}(\hat{Z}_m^\mu)^\top\left(\frac{\boldsymbol{W}_i \boldsymbol{x}_{im}^\mu}{\sqrt{D}}\right) + i\sum_{m=1}^{M} Z_m^{\mu\top}\hat{Z}_m^\mu\right\}$$

$$\times \prod_{j\neq i}^{D} \underbrace{\int_{\mathbb{R}^P} \mathrm{d}\boldsymbol{W}_j m_{j\to\mu}^t(\boldsymbol{W}_j) \exp\left\{-i\sum_{m=1}^{M}(\hat{Z}_m^\mu)^\top\left(\frac{\boldsymbol{W}_j \boldsymbol{x}_{jm}^\mu}{\sqrt{D}}\right)\right\}}_{\equiv I_j}\,,$$

(70)

It is time to define the mean and variance of the messages $m_{j\to\mu}^t$

$$\hat{\boldsymbol{W}}_{j\to\mu}^t \equiv \int_{\mathbb{R}^P} \mathrm{d}\boldsymbol{W}_j m_{j\to\mu}^t(\boldsymbol{W}_j)\boldsymbol{W}_j\,, \qquad \hat{C}_{j\to\mu}^t \equiv \int_{\mathbb{R}^P} \mathrm{d}\boldsymbol{W}_j m_{j\to\mu}^t(\boldsymbol{W}_j)\boldsymbol{W}_j\boldsymbol{W}_j^\top - \hat{\boldsymbol{W}}_{j\to\mu}^t(\hat{\boldsymbol{W}}_{j\to\mu}^t)^\top\,. \qquad (71)$$

In the limit $D\to\infty$ the term $I_j$ can be easily expanded. Then, we can recognize that the lowest terms in the expansion are nothing but simple functions of the messages (71), and obtain a resummed expression at leading order in $D$.

$$I_j = \int_{\mathbb{R}^P} \mathrm{d}\boldsymbol{W}_j m_{j\to\mu}^t(\boldsymbol{W}_j) e^{-i\sum_{m=1}^{M}(\hat{Z}_m^\mu)^\top\left(\frac{\boldsymbol{W}_j \boldsymbol{x}_{jm}^\mu}{\sqrt{D}}\right)} \approx \exp\left\{-i\sum_{m=1}^{M}\frac{\boldsymbol{x}_{jm}^\mu}{\sqrt{D}}(\hat{Z}_m^\mu)^\top\hat{\boldsymbol{W}}_{j\to\mu}^t + \sum_{m,m'=1}^{M}\frac{1}{2}\frac{\boldsymbol{x}_{jm}^\mu \boldsymbol{x}_{jm'}^\mu}{D}(\hat{Z}_m^\mu)^\top\hat{C}_{j\to\mu}^t\hat{Z}_{m'}^\mu\right\}\,,$$

$$\approx \exp\left\{-i\sum_{m=1}^{M}\frac{\boldsymbol{x}_{jm}^\mu}{\sqrt{D}}(\hat{Z}_m^\mu)^\top\hat{\boldsymbol{W}}_{j\to\mu}^t + \sum_{m}^{M}\frac{1}{2}\frac{(\boldsymbol{x}_{jm}^\mu)^2}{D}(\hat{Z}_m^\mu)^\top\hat{C}_{j\to\mu}^t\hat{Z}_m^\mu\right\}$$

and finally using the inverse Fourier transform, we obtain

$$\hat{m}_{\mu\to i}^t(\boldsymbol{W}_i) \simeq \frac{1}{\mathcal{Z}_{\mu\to i}}\frac{1}{(2\pi)^{PM}} \int_{\mathbb{R}^{P\times M}} \mathrm{d}Z\,\delta(y^\mu - g(Z^\mu)) \int_{\mathbb{R}^{P\times M}} \mathrm{d}\hat{Z}\,\exp\left\{-i\sum_{m=1}^{M}(\hat{Z}_m^\mu)^\top\left(\frac{\boldsymbol{W}_i \boldsymbol{x}_{im}^\mu}{\sqrt{D}}\right) + i\sum_{m=1}^{M}(Z_m^\mu)^\top\hat{Z}_m^\mu\right\}$$

$$\times \prod_{j\neq i}^{D} \exp\left\{-i\sum_{m=1}^{M}\frac{\boldsymbol{x}_{jm}^\mu}{\sqrt{D}}(\hat{Z}_m^\mu)^\top\hat{\boldsymbol{W}}_{j\to\mu}^t + \sum_{m}^{M}\frac{1}{2}\frac{(\boldsymbol{x}_{jm}^\mu)^2}{D}(\hat{Z}_m^\mu)^\top\hat{C}_{j\to\mu}^t\hat{Z}_m^\mu\right\}$$

$$= \frac{1}{\mathcal{Z}_{\mu\to i}}\frac{1}{(2\pi)^{PM}} \int_{\mathbb{R}^{P\times M}} \mathrm{d}Z\,\delta(y^\mu - g(Z^\mu)) \int_{\mathbb{R}^{P\times M}} \mathrm{d}\hat{Z}\prod_{m=1}^{M}\left[e^{i(\hat{Z}_m^\mu)^\top\left(Z_m^\mu - \frac{\boldsymbol{x}_{im}^\mu}{\sqrt{D}}\boldsymbol{W}_i - \sum_{j\neq i}^{D}\frac{\boldsymbol{x}_{jm}^\mu}{\sqrt{D}}\hat{\boldsymbol{W}}_{j\to\mu}^t\right) + \frac{(\boldsymbol{x}_{jm}^\mu)^2}{2D}(\hat{Z}_m^\mu)^\top\hat{C}_{j\to\mu}^t\hat{Z}_m^\mu}\right]$$

$$= \frac{1}{\mathcal{Z}_{\mu\to i}}\frac{1}{\sqrt{(2\pi)^{PM}\prod_{m=1}^{M}\det(V_{im}^{\mu,t})}} \int_{\mathbb{R}^{P\times M}} \mathrm{d}Z\,\delta(y^\mu - g(Z^\mu)) \prod_{m=1}^{M}\left[\underbrace{e^{-\frac{1}{2}\left(Z_m - \frac{\boldsymbol{x}_{im}^\mu}{\sqrt{D}}\boldsymbol{W}_i - \omega_{im}^{\mu,t}\right)^\top (V_{im}^{\mu,t})^{-1}\left(Z_m - \frac{\boldsymbol{x}_{im}^\mu}{\sqrt{D}}\boldsymbol{W}_i - \omega_{im}^{\mu,t}\right)}}_{\equiv H_{im}^\mu}\right]\,,$$

where we introduced the mean and variance of $H^\mu_{im}$

$$\omega^{\mu,t}_{im} \equiv \frac{1}{\sqrt{D}} \sum_{j \neq i}^{D} \boldsymbol{x}^\mu_{jm} \hat{W}^t_{j \to \mu}, \qquad V^{\mu,t}_{im} \equiv \frac{1}{D} \sum_{j \neq i}^{D} (\boldsymbol{x}^\mu_{jm})^2 \hat{C}^t_{j \to \mu}. \tag{72}$$

As we did, before we will expand the term $H^\mu_{im}$ in the $D \to \infty$ limit keeping the leading order terms

$$H^\mu_{im} \simeq e^{-\frac{1}{2}\left(Z_m - \omega^{\mu,t}_{im}\right)^\top (V^{\mu,t}_{im})^{-1}\left(Z_m - \omega^{\mu,t}_{im}\right)} \left(1 + \frac{\boldsymbol{x}^\mu_{im}}{\sqrt{D}} \boldsymbol{W}^\top_i (V^{\mu,t}_{im})^{-1}(Z_m - \omega^{\mu,t}_{im}) - \frac{1}{2}\frac{(\boldsymbol{x}^\mu_{im})^2}{D} \boldsymbol{W}^\top_i (V^{\mu,t}_{im})^{-1} \boldsymbol{W}_i \right.$$
$$\left. + \frac{1}{2}\frac{(\boldsymbol{x}^\mu_{im})^2}{D} \boldsymbol{W}^\top_i (V^{\mu,t}_{im})^{-1}(Z_m - \omega^{\mu,t}_{im})(Z_m - \omega^{\mu,t}_{im})^\top (V^{\mu,t}_{im})^{-1} \boldsymbol{W}_i \right).$$

As we resum the expression above, we can express it using $g_{\text{out}}$ and $\partial_\omega g_{\text{out}}$, as defined in (156). For convenience, we can define $g^{\mu,t} \in \mathbb{R}^P$, $G^{\mu,t}_m \in \mathbb{R}^{P \times P}$

$$g^{\mu,t}_m \equiv \left[g_{\text{out}}(y^\mu, \omega^{\mu,t}_i, V^{\mu,t}_i)\right]_m, \qquad G^{\mu,t}_m \equiv \left[\partial_\omega g_{\text{out}}(y^\mu, \omega^{\mu,t}_i, V^{\mu,t}_i)\right]_{mm}. \tag{73}$$

We arrive to the expression

$$\hat{m}^t_{\mu \to i}(\boldsymbol{W}_i) \approx \frac{1}{\mathcal{Z}_{\mu \to i}} \left\{1 + \sum_{m=1}^{M} \frac{\boldsymbol{x}^\mu_{im}}{\sqrt{D}} \boldsymbol{W}^\top_i g^{\mu,t}_m + \sum_{m=1}^{M} \frac{(\boldsymbol{x}^\mu_{im})^2}{2D} \boldsymbol{W}^\top_i g^{\mu,t}_m (g^{\mu,t}_m)^\top \boldsymbol{W}_i + \sum_{m=1}^{M} \frac{(\boldsymbol{x}^\mu_{im})^2}{2D} \boldsymbol{W}^\top_i G^{\mu,t}_i \boldsymbol{W}_i \right\}$$
$$= \frac{1}{\mathcal{Z}_{\mu \to i}} \left\{1 + \boldsymbol{W}^\top_i B^t_{\mu \to i} + \frac{1}{2}\boldsymbol{W}^\top_i B^t_{\mu \to i}(B^t_{\mu \to i})^\top \boldsymbol{W}_i - \frac{1}{2}\boldsymbol{W}^\top_i A^t_{\mu \to i} \boldsymbol{W}_i \right\} \tag{74}$$
$$\approx \sqrt{\frac{\det\left(A^t_{\mu \to i}\right)}{(2\pi)^P}} \exp \left\{-\frac{1}{2}\left(\boldsymbol{W}^\top_i - (A^t_{\mu \to i})^{-1}B^t_{\mu \to i}\right)^\top A^t_{\mu \to i}\left(\boldsymbol{W}^\top_i - (A^t_{\mu \to i})^{-1}B^t_{\mu \to i}\right)\right\},$$

where we implicitly defined $A_{\mu \to i}$ and $B_{\mu \to i}$:

$$B^t_{\mu \to i} \equiv \sum_{m=1}^{M} \frac{\boldsymbol{x}^\mu_{im}}{\sqrt{D}} g^{\mu,t}_m, \qquad A^t_{\mu \to i} \equiv -\sum_{m=1}^{M} \frac{(\boldsymbol{x}^\mu_{im})^2}{D} G^{\mu,t}_m. \tag{75}$$

The approximation in equation (74) can be plugged in the first equation of (68)

$$m^{t+1}_{i \to \mu}(\boldsymbol{W}_i) = \frac{1}{\mathcal{Z}_{i \to \mu}} \frac{e^{-\frac{\|\boldsymbol{w}_i\|^2}{2}}}{(2\pi)^{P/2}} \prod_{\nu \neq \mu}^{N} \sqrt{\frac{\det(A^t_{\nu \to i})}{(2\pi)^P}} \exp \left\{-\frac{1}{2}\left(\boldsymbol{W}^\top_i - (A^t_{\nu \to i})^{-1}B^t_{\nu \to i}\right)^\top A^t_{\nu \to i}\left(\boldsymbol{W}^\top_i - (A^t_{\nu \to i})^{-1}B^t_{\nu \to i}\right)\right\}$$
$$= \frac{1}{\mathcal{Z}_{i \to \mu}} \sqrt{\frac{\prod_{\nu \neq \mu}^{N} \det(A^t_{\nu \to i})}{(2\pi)^P}} e^{-\frac{1}{2}\boldsymbol{W}_i\left(\mathbb{1}_P + \sum_{\nu \neq \mu}^{N} A^t_{\nu \to i}\right)\boldsymbol{W}^\top_i + \boldsymbol{W}_i \sum_{\nu \neq \mu}^{N} B^t_{\nu \to i} - \frac{1}{2}\sum_{\nu \neq \mu}^{N} (B^t_{\nu \to i})^\top (A^t_{\nu \to i})^{-1} B^t_{\nu \to i}}. \tag{76}$$

We have come full circle, as we can now explicitly write the moments of $m^t_{i \to \mu}$ we first introduced in (71).

$$\hat{W}^{t+1}_{i \to \mu} = \left(\mathbb{1}_P + \sum_{\nu \neq \mu}^{N} A^t_{\nu \to i}\right)^{-1} \left(\sum_{\nu \neq \mu}^{N} B^t_{\nu \to i}\right), \qquad \hat{C}^{t+1}_{i \to \mu} = \left(\mathbb{1}_P + \sum_{\nu \neq \mu}^{N} A^t_{\nu \to i}\right)^{-1}. \tag{77}$$

What we have obtained is thus a consistent set of equations that iterates on the moments of the BP messages. We give a

compact description of this algorithm here

$$\omega_{im}^{\mu,t} = \frac{1}{\sqrt{D}} \sum_{j \neq i}^{D} \boldsymbol{x}_{jm}^{\mu} \hat{\boldsymbol{W}}_{j \to \mu}^{t}$$

$$V_{im}^{\mu,t} = \frac{1}{D} \sum_{j \neq i}^{D} (\boldsymbol{x}_{jm}^{\mu})^2 \hat{C}_{j \to \mu}^{t}$$

$$g_m^{\mu,t} = \left[ g_{\text{out}}(y^\mu, \omega_i^{\mu,t}, V_i^{\mu,t}) \right]_m$$

$$G_m^{\mu,t} = \left[ \partial_\omega g_{\text{out}}(y^\mu, \omega_i^{\mu,t}, V_i^{\mu,t}) \right]_{mm}$$

$$B_{\mu \to i}^{t} = \sum_{m=1}^{M} \frac{\boldsymbol{x}_{im}^{\mu}}{\sqrt{D}} g_m^{\mu,t} \tag{78}$$

$$A_{\mu \to i}^{t} = - \sum_{m=1}^{M} \frac{(\boldsymbol{x}_{im}^{\mu})^2}{D} G_m^{\mu,t}$$

$$\hat{\boldsymbol{W}}_{i \to \mu}^{t+1} = \left( \mathbb{1}_P + \sum_{\nu \neq \mu}^{N} A_{\nu \to i}^{t} \right)^{-1} \left( \sum_{\nu \neq \mu}^{N} B_{\nu \to i}^{t} \right),$$

$$\hat{C}_{i \to \mu}^{t+1} = \left( \mathbb{1}_P + \sum_{\nu \neq \mu}^{N} A_{\nu \to i}^{t} \right)^{-1}$$

## D.1. Relaxed BP implies the State Equation

A first result we can get from Relaxed BP is the state equation (14). The key ingredient to obtain that is noticing how at each step the quantities in (78) either concentrate or converge in distribution to Gaussian variables in the limit of large $D$.

The first step is to specify how $y^\mu$ is generated. We can do so by defining the index of the generative model $Z^\mu \in \mathbb{R}^{P \times M}$ and it is equivalent $Z_i^\mu$ with the weight $\boldsymbol{W}_j$ removed

$$Z_m^\mu = \frac{1}{\sqrt{D}} \sum_{j=1}^{D} \boldsymbol{x}_{jm}^\mu \boldsymbol{W}_j^*, \qquad\qquad Z_{im}^\mu = \frac{1}{\sqrt{D}} \sum_{j \neq i}^{D} \boldsymbol{x}_{jm}^\mu \boldsymbol{W}_j^*. \tag{79}$$

With this definition the labels are $y^\mu = g(Z^\mu)$. Now, in the limit of large $D$, $Z^\mu$ and $\omega^\mu$ are going to be jointly Gaussian at each step with the following covariance structure

$$\begin{bmatrix} Z_m^\mu \\ \omega_{im}^{\mu,t} \end{bmatrix} \sim \mathcal{N} \left( \begin{bmatrix} 0 \\ 0 \end{bmatrix}, \begin{bmatrix} \mathbb{1}_P & Q^t \\ Q^t & Q^t \end{bmatrix} \right), \tag{80}$$

where $Q^t$ is the same overlap of the weights estimated at time $t$ and the target

$$Q^t \equiv \mathbb{E} \left[ \lim_{D \to \infty} \frac{\hat{\boldsymbol{W}}_i^t (\boldsymbol{W}_i^*)^\top}{D} \right] = \mathbb{E} \left[ \lim_{D \to \infty} \frac{\hat{\boldsymbol{W}}_i^t (\hat{\boldsymbol{W}}_i^t)^\top}{D} \right], \tag{81}$$

where the second equality is because of the Nishimori identity. The expectation is taken over the target weights $\boldsymbol{W}^*$ and the input data $\boldsymbol{x}$. We can show this by looking at the first few moments of $Z_m^\mu$ and $\omega_{im}^{\mu,t}$

$$\mathbb{E}\left[ Z_{im}^\mu \right] = 0, \qquad \mathbb{E}\left[ Z_{im}^\mu (Z_{im}^\mu)^\top \right] = \mathbb{1}_P,$$

$$\mathbb{E}\left[ \omega_{im}^{\mu,t} \right] = 0, \qquad \mathbb{E}\left[ \omega_{im}^{\mu,t} (\omega_{im}^{\mu,t})^\top \right] = \mathbb{E}\left[ \frac{1}{D} \sum_{j \neq i}^{D} \hat{\boldsymbol{W}}_{j \to \mu}^t (\hat{\boldsymbol{W}}_{j \to \mu}^t)^\top \right] \approx \mathbb{E}\left[ \frac{1}{D} \sum_{j}^{D} \hat{\boldsymbol{W}}_j^t (\hat{\boldsymbol{W}}_j^t)^\top \right] = Q^t, \tag{82}$$

$$\mathbb{E}\left[ z_{im}^\mu (\omega_{im}^{\mu,t})^\top \right] = \mathbb{E}\left[ \frac{1}{D} \sum_{j \neq i}^{D} \boldsymbol{W}_j^* (\hat{\boldsymbol{W}}_{j \to \mu}^t)^\top \right] \approx \mathbb{E}\left[ \frac{1}{D} \sum_{j}^{D} \boldsymbol{W}_j^* (\hat{\boldsymbol{W}}_j^t)^\top \right] = Q^t.$$

Next, we derive the leading order behavior of $g_{\text{out}}$ and $\partial_\omega g_{\text{out}}$.

$$g_{\text{out}}(y^\mu, \omega_i^{\mu,t}, , V_i^{\mu,t}) = g_{\text{out}}(g(Z^{\mu,t}), \omega_i^{\mu,t}, , V_i^{\mu,t})$$

$$\approx g_{\text{out}}(g(Z_i^{\mu,t}), \omega_i^{\mu,t}, V_i^{\mu,t}) + \frac{1}{\sqrt{D}} \sum_{m=1}^M \boldsymbol{x}_{im}^\mu \partial_z \left[ g_{\text{out}}(g(Z_i^{\mu,t}), \omega_i^{\mu,t}, V_i^{\mu,t}) \right]_m \boldsymbol{W}_i^{*'} \tag{83}$$

$$\partial_\omega g_{\text{out}}(y^\mu, \omega_i^{\mu,t}, V_i^{\mu,t}) \approx \partial_\omega g_{\text{out}}(g(Z_i^{\mu,t}), \omega_i^{\mu,t}, V_i^{\mu,t}).$$

The covariances $V_{im}^{\mu,t}$ will concentrate

$$\mathbb{E}\left[V_{im}^{\mu,t}\right] \approx \frac{1}{D} \sum_{j=1}^D \mathbb{E}\left[\hat{C}_{j\to\mu}^t\right]$$

$$= \frac{1}{D} \sum_{j=1}^D \mathbb{E}\left[ \int_{\mathbb{R}^P} \mathrm{d}\boldsymbol{W}_j\, m_{j\to\mu}^{t-1}(\boldsymbol{W}_j) \boldsymbol{W}_j \boldsymbol{W}_j^\top \right] - \frac{1}{D} \sum_{j=1}^D \mathbb{E}\left[ \hat{\boldsymbol{W}}_{j\to\mu}^{t-1} (\hat{\boldsymbol{W}}_{j\to\mu}^{t-1})^\top \right] \tag{84}$$

$$= \mathbb{1}_P - Q^{t-1},$$

where in the first expectation on the second line we used Nishimori. We can then decompose the sum of the $B_{\nu\to i}^t$ as follows

$$\sum_{\nu\neq\mu}^N B_{\nu\to i}^t \approx \underbrace{\sum_{\mu,m}^{N,M} \frac{\boldsymbol{x}_{im}^\mu}{\sqrt{D}} g_{\text{out}}\left[g(Z_i^{\mu,t}), \omega_i^{\mu,t}, V_i^{\mu,t}\right]_m}_{\xi_i^t} + \underbrace{\frac{1}{D} \sum_{\mu,m}^{N,M} (x_{im}^\mu)^2 \left[\partial_z g_{\text{out}}(g(Z_i^{\mu,t}), \omega_i^{\mu,t}, V_i^{\mu,t}\right]_m \boldsymbol{W}_i^*}_{S^t \boldsymbol{W}_i^*}. \tag{85}$$

Here, $\xi_i^t \in \mathbb{R}^P$ is at each step a Gaussian variable with covariance $\hat{Q}^t$

$$\mathbb{E}[\xi_i^t] = 0, \qquad \mathbb{E}[\xi_i^t(\xi_i^t)^\top] = \hat{Q}^t, \tag{86}$$

where

$$\hat{Q}^t \equiv \alpha \mathbb{E}_{Z,\omega}\left[g_{\text{out}}(g(Z), \omega^t, \mathbb{1}_P - Q^t)\, g_{\text{out}}(g(Z), \omega^t, \mathbb{1}_P - Q^t)^\top\right], \tag{87}$$

and the expectation is taken over $Z$, $\omega$ distributed as in (80). It is possible to show with an extremely tedious computation that $S^t(S^t)^\top = \hat{Q}^t$. Finally, we have the State Equation

$$Q^{t+1} = \mathbb{E}\left[\frac{1}{D} \sum_{i=1}^D \hat{\boldsymbol{W}}_{i\to\mu}^{t+1}(\hat{\boldsymbol{W}}_{i\to\mu}^{t+1})^\top\right] = (1 + \hat{Q}^t)^{-1}\hat{Q}^t, \tag{88}$$

which is exactly the same expression as (14).

### D.2. The fixed points of the State Equation are extremisers of the the variational problem (10)

It is not immediate by just looking at it, but the State Evolution (14) at its fixed point describes the extremisers of the functional (10). To show this we first rewrite (10) as

$$\sup_{\hat{Q}\in\mathbb{S}_P^+} \inf_{Q\in\mathbb{S}_P^+} S(Q, \hat{Q}), \tag{89}$$

where

$$S(Q, \hat{Q}) = -\frac{1}{2}\text{Tr}(Q\hat{Q}) - \frac{1}{2}\log\left(\mathbb{1}_P + \hat{Q}\right) + \frac{1}{2}\hat{Q} + \alpha\mathbb{E}_{\xi\sim\mathcal{N}(0,\mathbb{1}_P)}\mathbb{E}_{z^0\sim\mathcal{N}(0,\mathbb{1}_P)}\log\mathcal{Z}_{\text{out}}\left(g(\sqrt{\mathbb{1}_P - Q}Z^0 + \sqrt{Q}\xi), \sqrt{Q}\xi, \mathbb{1}_P - Q\right). \tag{90}$$

where we already introduced for convenience $\mathcal{Z}_{\text{out}}$ and $g_{\text{out}}$, defined as

$$\mathcal{Z}_{\text{out}}(y, \omega, V) = \mathbb{E}_{\{z\sim\mathcal{N}(\omega,V)\}}\left[\delta(y - g(Z))\right], \qquad g_{\text{out}}(y, \omega, V) = \frac{\mathbb{E}_{\{z\sim\mathcal{N}(\omega,V)\}}\left[(Z - \omega)V^{-1}\delta(y - g(Z))\right]}{\mathbb{E}_{\{z\sim\mathcal{N}(\omega,V)\}}\left[\delta(y - g(Z))\right]}. \tag{91}$$

In the following we will use the identities

$$\partial_\omega \mathcal{Z}_{\text{out}} = \mathcal{Z}_{\text{out}} g_{\text{out}}\,, \qquad \partial_V \mathcal{Z}_{\text{out}} = \frac{1}{2} \mathcal{Z}_{\text{out}} (\partial_\omega g_{\text{out}} + g_{\text{out}} \otimes g_{\text{out}})\,. \tag{92}$$

The extremisers are going to satisfy the following system.

$$\begin{cases} \partial_Q S(Q,\hat{Q}) = 0 \\ \partial_{\hat{Q}} S(Q,\hat{Q}) = 0 \end{cases} \tag{93}$$

or equivalently

$$\begin{cases} \hat{Q} = 2\alpha \partial_Q \mathbb{E}_{\xi \sim \mathcal{N}(0,\mathbb{1}_P)} \mathbb{E}_{z^0 \sim \mathcal{N}(0,\mathbb{1}_P)} \log \mathcal{Z}_{\text{out}} \left(g(\sqrt{\mathbb{1}_P - Q} Z^0 + \sqrt{Q}\xi), \sqrt{Q}\xi, \mathbb{1}_P - Q\right) \\ Q = (\mathbb{1}_P - \hat{Q})^{-1} - \mathbb{1}_P = (\mathbb{1}_P - \hat{Q})^{-1}\hat{Q} \end{cases} \tag{94}$$

The hard part is computing the derivative with respect to $Q$ of the first equation above. We start by adding a dummy variable $y$, which allows us to have the identity

$$\partial_Q \mathbb{E}_\xi \mathbb{E}_{z^0} \log \mathcal{Z}_{\text{out}} \left(g(\sqrt{\mathbb{1}_P - Q} Z^0 + \sqrt{Q}\xi), \sqrt{Q}\xi, \mathbb{1}_P - Q\right) = $$
$$\partial_Q \int \mathrm{d}y\, \mathbb{E}_\xi \mathcal{Z}_{\text{out}} \left(y, \sqrt{Q}\xi, \mathbb{1}_P - Q\right) \log \mathcal{Z}_{\text{out}} \left(y, \sqrt{Q}\xi, \mathbb{1}_P - Q\right)\,. \tag{95}$$

We can now compute the derivative

$$\partial_Q \int \mathrm{d}y\, \mathbb{E}_\xi \mathcal{Z}_{\text{out}} \log \mathcal{Z}_{\text{out}}$$

$$= \int \mathrm{d}y\, \mathbb{E}_\xi \left[\partial_Q \mathcal{Z}_{\text{out}} \log \mathcal{Z}_{\text{out}} + \partial_Q \mathcal{Z}_{\text{out}}\right]$$

$$= \int \mathrm{d}y\, \mathbb{E}_\xi \left[\left(\frac{1}{2} Q^{-1/2}\xi \partial_\omega \mathcal{Z}_{\text{out}} - \partial_V \mathcal{Z}_{\text{out}}\right) \log \mathcal{Z}_{\text{out}} + \frac{1}{2} Q^{-1/2}\xi \partial_\omega \mathcal{Z}_{\text{out}} - \partial_V \mathcal{Z}_{\text{out}}\right]$$

$$= \int \mathrm{d}y\, \mathbb{E}_\xi \left[\left(\frac{1}{2} Q^{-1/2}\xi \mathcal{Z}_{\text{out}} g_{\text{out}} - \frac{1}{2}\mathcal{Z}_{\text{out}}(\partial_\omega g_{\text{out}} + g_{\text{out}} \otimes g_{\text{out}})\right) \log \mathcal{Z}_{\text{out}} \right.$$
$$\left. + \frac{1}{2} Q^{-1/2}\xi \mathcal{Z}_{\text{out}} g_{\text{out}} - \frac{1}{2}\mathcal{Z}_{\text{out}}(\partial_\omega g_{\text{out}} + g_{\text{out}} \otimes g_{\text{out}})\right]$$

$$= \frac{1}{2} \int \mathrm{d}y\, \mathbb{E}_\xi \left[\partial_\omega (\mathcal{Z}_{\text{out}} g_{\text{out}} \log \mathcal{Z}_{\text{out}}) - \mathcal{Z}_{\text{out}}(\partial_\omega g_{\text{out}} + g_{\text{out}} \otimes g_{\text{out}}) \log \mathcal{Z}_{\text{out}} + \partial_\omega (\mathcal{Z}_{\text{out}} g_{\text{out}}) - \mathcal{Z}_{\text{out}}(\partial_\omega g_{\text{out}} + g_{\text{out}} \otimes g_{\text{out}})\right]$$

$$= \frac{1}{2} \int \mathrm{d}y\, \mathbb{E}_\xi \left[\partial_\omega (\mathcal{Z}_{\text{out}} g_{\text{out}} \log \mathcal{Z}_{\text{out}}) - \mathcal{Z}_{\text{out}}(\partial_\omega g_{\text{out}} + g_{\text{out}} \otimes g_{\text{out}}) \log \mathcal{Z}_{\text{out}}\right]$$

$$= \frac{1}{2} \int \mathrm{d}y\, \mathbb{E}_\xi \left[\mathcal{Z}_{\text{out}} g_{\text{out}}^{\otimes 2}\right]\,, \tag{96}$$

where in the 5-th line we used Stein's lemma. Removing the dummy variable we get

$$\begin{cases} \hat{Q} = \alpha \mathbb{E}_{\xi \sim \mathcal{N}(0,\mathbb{1}_P)} \mathbb{E}_{z^0 \sim \mathcal{N}(0,\mathbb{1}_P)} g_{\text{out}} \left(g(\sqrt{\mathbb{1}_P - Q} Z^0 + \sqrt{Q}\xi), \sqrt{Q}\xi, \mathbb{1}_P - Q\right)^{\otimes 2} \\ Q = (\mathbb{1}_P - \hat{Q})^{-1}\hat{Q} \end{cases} \tag{97}$$

which is equivalent to the state equation without time indices.

**D.3. From relaxed BP to GAMP**

The relaxed BP equations (78) describe a perfectly usable algorithm with $\mathcal{O}(D^2)$ time complexity. We can do better and obtain a linear time iterative scheme. Let us define the following quantities

$$
\begin{aligned}
\omega_m^{\mu,t} &= \frac{1}{\sqrt{D}} \sum_{j=1}^{D} \boldsymbol{x}_{jm}^{\mu} \hat{\boldsymbol{W}}_{j\to\mu}^{t} \,, \\
V_m^{\mu,t} &= \frac{1}{D} \sum_{j=1}^{D} (\boldsymbol{x}_{jm}^{\mu})^2 \hat{C}_{j\to\mu}^{t} \,, \\
\hat{\boldsymbol{W}}_i^{t+1} &= \left( \mathbb{1}_P + \sum_{\nu=1}^{N} A_{\nu\to i}^{t} \right)^{-1} \left( \sum_{\nu=1}^{N} B_{\nu\to i}^{t} \right) \,, \\
\hat{C}_i^{t+1} &= \left( \mathbb{1}_P + \sum_{\nu=1}^{N} A_{\nu\to i}^{t} \right)^{-1} \,,
\end{aligned}
\tag{98}
$$

which are very similar to their equivalent in relaxed BP. The idea is to have a set of consistent equations on these new quantities. Since here the moments $\omega, V, \hat{W}, \hat{C}$ have one less index that in relaxed BP, the algorithm will be $\mathcal{O}(D)$ instead of $\mathcal{O}(D^2)$.

For large $D$ we have that at leading order $\hat{C}_{i\to\mu}^{t+1} \approx \hat{C}_i^{t+1}$. Similarly, $V_m^{\mu}$ is changed minimally.

$$
V_m^{\mu,t} \approx \frac{1}{D} \sum_{j=1}^{D} (\boldsymbol{x}_{jm}^{\mu})^2 \hat{C}_j^{t} \,.
\tag{99}
$$

The other quantities are a bit more delicate. First, notice how $\hat{w}_i$ has a non-negligible correction of order $\mathcal{O}(1/\sqrt{D})$

$$
\begin{aligned}
\hat{\boldsymbol{W}}_{i\to\mu}^{t+1} &\approx \left( \mathbb{1}_P + \sum_{\nu=1}^{N} A_{\nu\to i}^{t} \right)^{-1} \left( \sum_{\nu=1}^{N} B_{\nu\to i}^{t} - \sum_{m=1}^{M} \frac{\boldsymbol{x}_{im}^{\mu}}{\sqrt{D}} \left[ g_{\text{out}}(y^{\mu}, \omega_i^{\mu,t}, V_i^{\mu,t}) \right]_m \right) \\
&\approx \hat{\boldsymbol{W}}_i^{t+1} - \hat{C}_i^{t+1} \sum_{m=1}^{M} \frac{\boldsymbol{x}_{im}^{\mu}}{\sqrt{D}} \left[ g_{\text{out}}(y^{\mu}, \omega_i^{\mu,t}, V_i^{\mu,t}) \right]_m \,.
\end{aligned}
\tag{100}
$$

Similarly, removing one index to $\omega$ and $V$ inside $g_{\text{out}}$ produces an extra correction in $\mathcal{O}(1/\sqrt{D})$

$$
g_{\text{out}}(y^{\mu}, \omega_i^{\mu,t}, V_i^{\mu,t}) \approx g_{\text{out}}(y^{\mu}, \omega^{\mu,t}, V^{\mu,t}) - \sum_{m=1}^{M} \frac{\boldsymbol{x}_{im}^{\mu}}{\sqrt{D}} \left[ \partial_\omega g_{\text{out}}(y^{\mu}, \omega^{\mu,t}, V^{\mu,t}) \right]_m \hat{\boldsymbol{W}}_i^{t} \,.
\tag{101}
$$

All this results in a correction in the equation for $\omega$, typically called Onsager reaction term

$$
\omega_m^{\mu,t} \approx \frac{1}{\sqrt{D}} \sum_{j=1}^{D} \boldsymbol{x}_{jm}^{\mu} \hat{\boldsymbol{W}}_j^{t} - \frac{1}{D} \sum_{j=1}^{D} (\boldsymbol{x}_{jm}^{\mu})^2 \hat{C}_i^{t-1} \left[ g_{\text{out}}(y^{\mu}, \omega^{\mu,t-1}, V^{\mu,t-1}) \right]_m \,.
\tag{102}
$$

The moment $B_{\mu\to i}$ similarly receives a correction

$$
B_{\mu\to i}^{t} \approx \sum_{m=1}^{M} \frac{\boldsymbol{x}_{im}^{\mu}}{\sqrt{D}} \left[ g_{\text{out}}(y^{\mu}, \omega^{\mu,t}, V^{\mu,t}) \right]_m - \frac{1}{D} \sum_{m=1}^{M} (\boldsymbol{x}_{im}^{\mu})^2 \left[ \partial_\omega g_{\text{out}}(y^{\mu}, \omega^{\mu,t}, V^{\mu,t}) \right]_{mm} \hat{\boldsymbol{W}}_i^{t} \,.
\tag{103}
$$

It is convenient to define two new quantities $A^t$ and $b_i^t$

$$
\begin{aligned}
A_i^t &\equiv \sum_{\nu=1}^{N} A_{\mu\to i}^{t} \approx -\frac{1}{D} \sum_{\mu,m}^{N,M} (\boldsymbol{x}_{im}^{\mu})^2 \left[ \partial_\omega g_{\text{out}}(y^{\mu}, \omega^{\mu,t}, V^{\mu,t}) \right]_{mm} \,, \\
b_i^t &\equiv \sum_{\nu=1}^{N} B_{\mu\to i}^{t} \approx \sum_{\mu,m}^{N,M} \frac{\boldsymbol{x}_{im}^{\mu}}{\sqrt{D}} \left[ g_{\text{out}}(y^{\mu}, \omega^{\mu,t}, V^{\mu,t}) \right]_m + A_i^t \hat{\boldsymbol{W}}_i^{t} \,,
\end{aligned}
\tag{104}
$$

from which we finally obtain

$$\hat{\boldsymbol{W}}_i^{t+1} = \left(\mathbb{1}_P + A_i^t\right)^{-1} b_i^t,$$
$$\hat{C}_i^{t+1} = \left(\mathbb{1}_P + A_i^t\right)^{-1}, \tag{105}$$

This concludes the derivation. At very large $D$, it is possible to introduce $\hat{C}^t$, $V^t$, $A^t$ as the version without indices of $\hat{C}_i^t$, $V_m^{\mu,t}$, $A_i^t$

$$\hat{C}^t = \left(\mathbb{1}_P + A^t\right)^{-1}$$
$$A^t = -\frac{\alpha}{N} \sum_{\mu,m}^{N,M} \left[\partial_\omega g_{\text{out}}(y^\mu, \omega^{\mu,t}, V^{\mu,t})\right]_{mm} \tag{106}$$
$$V^t = \hat{C}^t.$$

This simplified algorithm is described in pseudo-code in Algorithm 1 in the main text.

### D.4. AMP with side information on the target

In this section we generalize GAMP to include some side information on the weights $\boldsymbol{W}^*$ used to generate the data. Suppose we are provided not only with the dataset $\mathcal{D} = \{\boldsymbol{x}^\mu, y^\mu\}_{\mu=1}^N$, but also with a side information matrix $S \in \mathbb{R}^{P \times D}$

$$S = \sqrt{\lambda}\boldsymbol{W}^* + \sqrt{1-\lambda}\zeta, \tag{107}$$

where $\zeta \sim \mathcal{N}(0, \mathbb{1})$ is distributed as a standard Gaussian variable and $\lambda > 0$ is a small constant. This additional information can be encoded in the prior on the weights $w$

$$\boldsymbol{W}_i \sim \frac{1}{(2\pi(1-\lambda))^{P/2}} \exp\left\{-\frac{\|\boldsymbol{W}_i - \sqrt{\lambda}S_i\|^2}{2(1-\lambda)}\right\}. \tag{108}$$

We can now go back and go through the derivation again with this new prior. Eq. (76) becomes

$$m_{i\to\mu}^{t+1}(\boldsymbol{W}_i) = \frac{1}{\mathcal{Z}_{i\to\mu}} \sqrt{\frac{\prod\limits_{\nu\neq\mu}^N \det(A_{\nu\to i}^t)}{(2\pi(1-\lambda))^P}}\, e^{-\frac{1}{2}\boldsymbol{W}_i\left(\frac{1}{1-\lambda}\mathbb{1}_P + \sum\limits_{\nu\neq\mu}^N A_{\nu\to i}^t\right)\boldsymbol{W}_i^\top + \boldsymbol{W}_i\left(\frac{\sqrt{\lambda}}{1-\lambda}S_i + \sum\limits_{\nu\neq\mu}^N B_{\nu\to i}^t\right) - \frac{1}{2}\sum\limits_{\nu\neq\mu}^N (B_{\nu\to i}^t)^\top (A_{\nu\to i}^t)^{-1} B_{\nu\to i}^t}, \tag{109}$$

which gives us the new relaxed BP equations at leading order in small $\lambda$

$$\hat{\boldsymbol{W}}_{i\to\mu}^{t+1} = \left((1-\lambda)\mathbb{1}_P + \sum_{\nu\neq\mu}^N A_{\nu\to i}^t\right)^{-1}\left((1-\lambda)\sum_{\nu\neq\mu}^N B_{\nu\to i}^t + \sqrt{\lambda}S_i\sum_{\nu\neq\mu}^N A_{\nu\to i}^t\right), \quad \hat{C}_{i\to\mu}^{t+1} = \left((1-\lambda)\mathbb{1}_P + \sum_{\nu\neq\mu}^N A_{\nu\to i}^t\right)^{-1}, \tag{110}$$

and their equivalent in GAMP

$$\hat{\boldsymbol{W}}_i^{t+1} = \left((1-\lambda)\mathbb{1}_P + A_i^t\right)^{-1}\left((1-\lambda)b_i^t + \sqrt{\lambda}S_i A_i^t\right), \qquad \hat{C}_i^{t+1} = \left((1-\lambda)\mathbb{1}_P + A_i^t\right)^{-1}. \tag{111}$$

The state equation (88) is then slightly modified

$$Q^{t+1} = (1 + (1-\lambda)\hat{Q}^t)^{-1}((1-\lambda)\hat{Q}^t + \lambda). \tag{112}$$

### D.5. Sketch of the derivation of the State Evolution using the replica method

In this section, we provide a sketch of how to characterize the performance of a Bayes Optimal estimation using the replica method, a heuristic procedure inspired by the Statistical Physics of Disordered Systems (Gardner & Derrida, 1988; Mézard et al., 1987; Mezard & Montanari, 2009). Despite being non-rigorous, the method is very flexible and often provides a result

that can be later proven rigorously (Barbier et al., 2019; Aubin et al., 2018). We invite the interested reader to consult the many reviews available for a more detailed explanation (Zdeborová & Krzakala, 2016; Lauditi et al., 2024). A closely related characterization of the learning of SMI models using the replica method was first presented in (Cui, 2025) for empirical risk minimization problems.

Let us consider a generic sequence multi-index model. We are interested in studying a model with $P$ indices $Z_p$, each describing sequences of length $M$. By this we mean that we have a dataset $\mathcal{D} = \{\boldsymbol{x}^\mu, y^\mu\}_{\mu=1}^N$ of $N$ labeled samples. Each sample $\boldsymbol{x}^\mu$ is a sequence of $M$ tokens of dimension $D$ taken from a standard Gaussian distribution:

$$\boldsymbol{x}^\mu \in \mathbb{R}^{D \times M}, \qquad \boldsymbol{x}_{i,m}^\mu \sim \mathcal{N}(0,1). \tag{113}$$

The indices are obtained by projecting the input data onto a set of $P$ weight vectors $\boldsymbol{w}_p$ with standard Gaussian entries

$$\boldsymbol{w}_p \in \mathbb{R}^D, \qquad \boldsymbol{w}_{p,i}^* \sim \mathcal{N}(0,1), \qquad Z_p^\mu = \frac{\boldsymbol{w}_p^* x^\mu}{\sqrt{D}}. \tag{114}$$

In the following, we will call $\boldsymbol{W} \in \mathbb{R}^{P \times D}$ the weight matrix obtained by stacking the vectors $\boldsymbol{w}_p$. The labels are generated by linking together the indices with thorough a function $g$ such that

$$g : \mathbb{R}^{P \times M} \to \mathbb{R}^{M'}, \qquad y_\mu = g(Z_1^\mu, ..., Z_P^\mu). \tag{115}$$

Our results will hold in the high dimensional, proportional limit, that is when $N, D \to \infty$ with a finite ratio $\alpha = N/D$. All the other quantities $M$, $M'$, and $P$ are assumed to be much smaller than $D$. We are interested in characterizing the performance of a Bayes Optimal (BO) estimator on average over the dataset. The derivation starts with Bayes theorem:

$$\mathbb{P}(\boldsymbol{W}|\mathcal{D}) = \frac{\mathbb{P}(\boldsymbol{W}) \prod_{\mu=1}^N \mathbb{P}(y^\mu|\boldsymbol{W}, \boldsymbol{x}^\mu)}{\mathbb{P}(\mathcal{D})} = \frac{\mathbb{P}(\boldsymbol{W})}{\mathcal{Z}} \prod_{\mu=1}^N \delta\left(y^\mu - g\left(\frac{\boldsymbol{w}_1 \boldsymbol{x}^\mu}{\sqrt{D}}, ..., \frac{\boldsymbol{w}_P \boldsymbol{x}^\mu}{\sqrt{D}}\right)\right). \tag{116}$$

where $\mathcal{Z}$ is a normalization constant

$$\mathcal{Z} = \mathbb{E}_{\boldsymbol{W}}\left[\prod_{\mu=1}^N \delta\left(y^\mu - g\left(\frac{\boldsymbol{w}_1 \boldsymbol{x}^\mu}{\sqrt{D}}, ..., \frac{\boldsymbol{w}_P \boldsymbol{x}^\mu}{\sqrt{D}}\right)\right)\right]. \tag{117}$$

On the other hand, what we are interested in is the BO estimator *on average over the randomness of the dataset*, To this purpose,e we are resolved to compute the averaged cumulant generating function (or free entropy):

$$\phi = \lim_{D \to \infty} \frac{1}{D} \mathbb{E}_\mathcal{D}[\log \mathcal{Z}] = \lim_{n \to 0} \frac{\partial}{\partial n} \lim_{D \to \infty} \frac{1}{D} \log \mathbb{E}_\mathcal{D}[\mathcal{Z}^n], \tag{118}$$

where the last equality is called the replica trick: we consider a system with $n \in \mathbb{N}^+$ copies (or replicas) of the original model, extract the leading behaviour at large $D$, then analytically prolong the result to real valued $n$ and take the limit $n \to 0$. We are thus allowed to simply compute the moments of $\mathcal{Z}$

$$\mathbb{E}_\mathcal{D}[\mathcal{Z}^n] = \mathbb{E}_{\mathcal{D}, \boldsymbol{W}^a}\left[\prod_{\mu, a=1}^{N,n} \delta\left(y_\mu - g\left(\frac{\boldsymbol{w}_1^a \boldsymbol{x}^\mu}{\sqrt{D}}, ..., \frac{\boldsymbol{w}_P^a \boldsymbol{x}^\mu}{\sqrt{D}}\right)\right)\right]. \tag{119}$$

To make the expression more symmetric we think of the labels $y^\mu$ as generated by a zeroth replica

$$y^\mu = g\left(\frac{\boldsymbol{w}_1^0 \boldsymbol{x}^\mu}{\sqrt{D}}, ..., \frac{\boldsymbol{w}_P^0 \boldsymbol{x}^\mu}{\sqrt{D}}\right), \tag{120}$$

giving us

$$\mathbb{E}_\mathcal{D} \mathcal{Z}^n = \mathbb{E}_{\boldsymbol{x}^\mu, \boldsymbol{W}^a}\left[\prod_{\mu, a=1}^{N,n} \delta\left(g\left(\frac{\boldsymbol{w}_1^0 \boldsymbol{x}^\mu}{\sqrt{D}}, ..., \frac{\boldsymbol{w}_P^0 \boldsymbol{x}^\mu}{\sqrt{D}}\right) - g\left(\frac{\boldsymbol{w}_1^a \boldsymbol{x}^\mu}{\sqrt{D}}, ..., \frac{\boldsymbol{w}_P^a \boldsymbol{x}^\mu}{\sqrt{D}}\right)\right)\right]. \tag{121}$$

Now notice that all the indices are jointly correlated Gaussian variables:

$$Z_p^{a,\mu} = \frac{\boldsymbol{w}_p^a \boldsymbol{x}^\mu}{\sqrt{D}} \in \mathbb{R}^M \,, \qquad \mathbb{E}Z_p^{a,\mu} = 0 \,, \quad \mathbb{E}Z_{p,m}^{a,\mu} Z_{q,l}^{b,\nu} = Q_{pq}^{ab}\delta_{\mu\nu}\delta_{ml} \,, \tag{122}$$

where we introduced the overlaps $Q_{pq}^{ab}$:

$$Q_{pq}^{ab} = \frac{1}{D}\sum_{i=1}^D \boldsymbol{w}_{i,p}^a \boldsymbol{w}_{i,q}^b \,. \tag{123}$$

Since $Q_{pq}^{ab}$ contains also the covariance of the target weights $\boldsymbol{W}^*$ we have $Q_{pq}^{00} = \delta_{pq}$. Our goal will be to show that the overlaps $Q_{pq}^{ab}$ are the only quantity that characterizes the model, and that the partition function can be written a supremum over them. With these new definitions, we can rewrite the partition function as

$$\mathbb{E}_{\mathcal{D}}\mathcal{Z}^n = \int \mathrm{d}Q_{pq}^{ab}\, I_{\mathrm{prior}}(\{Q_{pq}^{ab}\})I_{\mathrm{channel}}(\{Q_{pq}^{ab}\})^N \,, \tag{124}$$

where the quantities $I_{\mathrm{prior}}$ and $I_{\mathrm{channel}}$ are pieces that depend only on the prior or the channel respectively:

$$I_{\mathrm{prior}} = \mathbb{E}_{\boldsymbol{W}^a}\left[\prod_{p,q=1,a,b=0}^{P,n} \delta\left(DQ_{pq}^{ab} - \boldsymbol{w}_p^a \boldsymbol{w}_q^b\right)\right] \,, \tag{125}$$

$$I_{\mathrm{channel}} = \mathbb{E}_{z_p^a \sim \mathcal{N}(0,Q)}\left[\prod_{a=1}^n \delta\left(g\left(Z_1^0, ..., Z_P^0\right) - g\left(Z_1^a, ..., Z_P^a\right)\right)\right] \,. \tag{126}$$

In the second expression we introduced a new variable $z_p^a \sim \mathcal{N}(0,Q)$ as the distribution of each of the tokens of $Z_p^a$, that is $z_p^a \sim Z_{pm}^a$. We can massage these two quantities into compact expressions. We start with $I_{\mathrm{prior}}$:

$$I_{\mathrm{prior}} = \mathbb{E}_{\boldsymbol{W}^a}\left[\prod_{p,q=1,a,b=0}^{P,n} \delta\left(DQ_{pq}^{ab} - \boldsymbol{w}_p^a(\boldsymbol{w}_q^b)^\top\right)\right] \tag{127}$$

$$= \mathbb{E}_{\boldsymbol{W}^a}\left[\exp\left\{\sum_{p,q=1,a,b=0}^{P,n} D\hat{Q}_{pq}^{ab}Q_{pq}^{ab} - \sum_{p,q=1,a,b=0}^{P,n}\hat{Q}_{pq}^{ab}\boldsymbol{w}_p^a(\boldsymbol{w}_q^b)^\top\right\}\right] \tag{128}$$

$$= \mathbb{E}_{\boldsymbol{W}^a}\left[\exp\left\{\sum_{p,q=1,a,b=0}^{P,n} D\hat{Q}_{pq}^{ab}Q_{pq}^{ab} - \sum_{p,q=1,a,b=0}^{P,n}\hat{Q}_{pq}^{ab}\sum_{i=1}^D \boldsymbol{w}_{i,p}^a\boldsymbol{w}_{i,q}^b\right\}\right] \tag{129}$$

$$= \exp\left\{\sum_{p,q=1,a,b=0}^{P,n} D\hat{Q}_{pq}^{ab}Q_{pq}^{ab}\right\}\mathbb{E}_{\boldsymbol{W}^a}\left[\exp\left\{-\sum_{p,q=1,a,b=0}^{P,n}\hat{Q}_{pq}^{ab}\boldsymbol{w}_p^a\boldsymbol{w}_q^b\right\}\right]^D \tag{130}$$

$$= \exp\left\{\sum_{p,q=1,a,b=0}^{P,n} D\hat{Q}_{pq}^{ab}Q_{pq}^{ab} - \frac{D}{2}\log\det\left(2\hat{Q}_{pq}^{ab}\right)\right\} \,. \tag{131}$$

Since we are only interested in the large $D$ behaviour, we can use the saddle point approximation

$$\left(Q_{pq}^{ab}\right)^{-1} = 2\hat{Q}_{pq}^{ab} \,, \tag{132}$$

which gives us the simple expression

$$I_{\mathrm{prior}} = \exp\left\{\frac{D}{2}\log\det\left(Q_{pq}^{ab}\right)\right\} \,. \tag{133}$$

This is a good moment to state some properties of the overlap $Q$. First, as we consider $n+1$ identical replicas in the model, $Q_{pq}^{ab}$ needs to be symmetric under exchange of replicas $a \leftrightarrow b$, commonly called the Nishimori condition (Nishimori, 1980). We then make a "replica symmetric" ansatz for the overlap

$$Q_{pq}^{ab} = \delta_{ab}\delta_{pq} + (1-\delta_{ab})Q_{pq} \,. \tag{134}$$

which simplifies the equations and allows us to analytically continue the theory. It is similarly reasonable to assume that $Q_{pq}$ are symmetric, positive definite matrices: $Q_{pq} \in \mathcal{S}_P^+$. After a tedious computation, it is possible to show the following identities

$$det(Q_{pq}^{ab}) = \det(\mathbb{1}_P - Q)^n \det(\mathbb{1}_P + nQ), \tag{135}$$

$$(Q^{-1})^{aa} = (\mathbb{1}_P - Q)^{-1}(\mathbb{1}_P + (n-1)Q)(\mathbb{1}_P + nQ)^{-1}, \tag{136}$$

$$(Q^{-1})^{ab} = -(\mathbb{1}_P - Q)^{-1}Q(\mathbb{1}_P + nQ)^{-1}. \tag{137}$$

The quantity $I_{\mathrm{prior}}$ can now be simplified even further

$$I_{\mathrm{prior}} = \exp\left\{\frac{D}{2}\left[n\log\det(\mathbb{1}_P - Q) + \log\det(\mathbb{1}_P + nQ)\right]\right\} \approx \exp\left\{\frac{nD}{2}\left[\log\det(\mathbb{1}_P - Q) + \mathrm{Tr}\,Q\right]\right\}, \tag{138}$$

where the approximation is valid for small enough $n$.

We can now focus on the channel part $I_{\mathrm{channel}}$ of the partition function. We will show how the distribution over the variables $z_p^a$ assumes a nice expression in the $n \to 0$ limit:

$$\frac{\mathrm{d}z}{\det Q}exp\left\{-\frac{1}{2}\sum_{p,q=1,a,b=0}^{P,n} z_p^a(Q^{-1})_{pq}^{ab}z_q^b\right\} \tag{139}$$

$$= \frac{\mathrm{d}z}{\det Q}exp\left\{-\frac{1}{2}\sum_{p,q=1,a=0}^{P,n} z_p^a(Q^{-1})_{pq}^{00}z_q^a - \frac{1}{2}\sum_{p,q=1,a\neq b}^{P,n} z_p^a(Q^{-1})_{pq}^{01}z_q^b\right\} \tag{140}$$

$$= \frac{\mathrm{d}z}{\det Q}exp\left\{-\frac{1}{2}\sum_{p,q=1,a=0}^{P,n} z_p^a[(Q^{-1})_{pq}^{00} - (Q^{-1})_{pq}^{01}]z_q^a - \frac{1}{2}\sum_{p,q=1,a,b=0}^{P,n} z_p^a(Q^{-1})_{pq}^{01}z_q^b\right\} \tag{141}$$

$$= \frac{\mathrm{d}z\mathrm{d}\xi}{\det Q}exp\left\{-\frac{1}{2}\sum_{p,q=1,a=0}^{P,n} z_p^a[(Q^{-1})_{pq}^{00} - (Q^{-1})_{pq}^{01}]z_q^a - i\sum_{p,q=1,a=0}^{P,n}\xi_p\sqrt{(Q^{-1})_{pq}^{01}}z_q^a - \frac{1}{2}\sum_{p=1}^{P}\xi_p^2\right\} \tag{142}$$

In the limit $n \to 0$ the identities (135) become

$$det(Q_{pq}^{ab}) \to 1, \tag{143}$$

$$(Q^{-1})^{aa} \to \mathbb{1}_P, \tag{144}$$

$$(Q^{-1})^{ab} = -(\mathbb{1}_P - Q)^{-1}Q. \tag{145}$$

As we can see in (142), it is possible to write a factorized distribution on each replica of $z_p$, while all of them are coupled with an external random variable $\xi$. We will drop the replica index $a$ to have:

$$\frac{\mathrm{d}z\mathrm{d}\xi}{\det Q}exp\left\{-\frac{1}{2}\sum_{p,q=1,a,b=0}^{P,n} z_p^a(Q^{-1})_{pq}^{ab}z_q^b\right\} = \mathrm{d}\xi e^{-\frac{1}{2}\|\xi\|^2}\left[\mathrm{d}z\,exp\left\{-\frac{1}{2}z^\top(\mathbb{1}_P - Q)^{-1}z + \xi^\top\sqrt{(\mathbb{1}_P - Q)^{-1}Q}z\right\}\right]^{n+1}. \tag{146}$$

We can now do a change of variable $\zeta \to \sqrt{\mathbb{1}_P - Q}\zeta$ and massage the result to obtain

$$\frac{\mathrm{d}z\mathrm{d}\xi}{\det Q}exp\left\{-\frac{1}{2}\sum_{p,q=1,a,b=0}^{P,n} z_p^a(Q^{-1})_{pq}^{ab}z_q^b\right\} = \mathrm{d}\xi e^{-\frac{1}{2}\|\xi\|^2}\left[\frac{\mathrm{d}z}{\sqrt{\det(\mathbb{1}_P - Q)}}exp\left\{-\frac{(z - \sqrt{Q}\xi)^\top(\mathbb{1}_P - Q)^{-1}(z - \sqrt{Q}\xi)}{2}\right\}\right]^{n+1}. \tag{147}$$

The computation is almost done: define $\mathcal{Z}_{\mathrm{out}}(y, \omega, V)$ as

$$\mathcal{Z}_{\mathrm{out}}(y, \omega, V) = \mathbb{E}_{\{z \sim \mathcal{N}(\omega, V)\}}[\delta(y - g(Z))] \tag{148}$$

from which we get for small $n$

$$I_{\text{channel}} = \mathbb{E}_{\xi \sim \mathcal{N}(0,\mathbb{1}_P)} \mathbb{E}_{z^0 \sim \mathcal{N}(\sqrt{Q}\xi, \mathbb{1}_P - Q)} \left[ \mathbb{E}_{z \sim \mathcal{N}(\sqrt{Q}\xi, \mathbb{1}_P - Q)} [\delta\left(g(Z^0) - g(Z)\right)] \right]^n \tag{149}$$

$$= \mathbb{E}_{\xi \sim \mathcal{N}(0,\mathbb{1}_P)} \mathbb{E}_{z^0 \sim \mathcal{N}(\sqrt{Q}\xi, \mathbb{1}_P - Q)} \mathcal{Z}_{\text{out}} \left( g(Z^0), \sqrt{Q}\xi, \mathbb{1}_P - Q \right)^n \tag{150}$$

$$\approx 1 + n \mathbb{E}_{\xi \sim \mathcal{N}(0,\mathbb{1}_P)} \mathbb{E}_{Z^0 \sim \mathcal{N}(\sqrt{Q}\xi, \mathbb{1}_P - Q)} \log \mathcal{Z}_{\text{out}} \left( g(Z^0), \sqrt{Q}\xi, \mathbb{1}_P - Q \right) \tag{151}$$

$$\approx \exp\left\{ \left\{ n \mathbb{E}_{\xi \sim \mathcal{N}(0,\mathbb{1}_P)} \mathbb{E}_{z^0 \sim \mathcal{N}(\sqrt{Q}\xi, \mathbb{1}_P - Q)} \log \mathcal{Z}_{\text{out}} \left( g(Z^0), \sqrt{Q}\xi, \mathbb{1}_P - Q \right) \right\} \right\}. \tag{152}$$

Bringing everything together we have

$$\frac{1}{nD} \log \mathbb{E}_{\mathcal{D}} \mathcal{Z}^n = S(Q) = \frac{1}{2} \left[ \log \det(\mathbb{1}_P - Q) + \text{Tr}\, Q \right] + \alpha \mathbb{E}_{\xi \sim \mathcal{N}(0,\mathbb{1}_P)} \mathbb{E}_{z^0 \sim \mathcal{N}(\sqrt{Q}\xi, \mathbb{1}_P - Q)} \log \mathcal{Z}_{\text{out}} \left( g(Z^0), \sqrt{Q}\xi, \mathbb{1}_P - Q \right) \tag{153}$$

Since we are interested in the large $D$ limit, we can use the saddle point approximation to obtain the free entropy:

$$\phi = \sup_{Q \in \mathcal{S}_P^+} S(Q). \tag{154}$$

The supremum $\tilde{Q}$ is achieved for $\partial_Q S(\tilde{Q}) = 0$. After a tedious computation it is possible to show that the supremum is achieved at the solution of the following fixed point equation, typically called the state equation:

$$\tilde{Q} = F \left[ \alpha \mathbb{E}_{Z^0, \xi} \left[ g_{\text{out}} \left( g(Z^0), \sqrt{\tilde{Q}}\xi, \mathbb{1}_P - Q \right) g_{\text{out}} \left( g(Z^0), \sqrt{\tilde{Q}}\xi, \mathbb{1}_P - \tilde{Q} \right)^\top \right] \right] \tag{155}$$

with $F[X] = (\mathbb{1}_P - X)^{-1} X$, $Z^0 \sim \sqrt{\mathbb{1}_P - \tilde{Q}} Z + \sqrt{\tilde{Q}}\xi$, $Z \sim \mathcal{N}(0, \mathbb{1}_P)$. Lastly, $g_{\text{out}}$ is:

$$g_{\text{out}}(y, \omega, V) = \frac{\mathbb{E}_{\{z \sim \mathcal{N}(\omega, V)\}} \left[ (Z - \omega) V^{-1} \delta(y - g(Z)) \right]}{\mathbb{E}_{\{z \sim \mathcal{N}(\omega, V)\}} \left[ \delta(y - g(Z)) \right]} \tag{156}$$

## E. Specialization to deep attention models

The main theoretical results of subsection 2.1 are stated for the broad class of SMI functions (1). These results are formulated, in particular, in terms of the denoiser function $g_{\text{out}}$ (12), whose form is tributary to the specific architecture considered. In this Appendix, we derive the expression of $g_{\text{out}}$ for the special case of rank $P_l = 1$ deep attention functions and length $M = 2$ data – thereby specializing the characterizations of Theorem 2.3 and Lemma 2.2 to this specific architecture. This specialization allows on the one hand to plug the specialized $g_{\text{out}}$ in Algorithm 1 and thus implement AMP for deep attention networks, and on the other and to reach the tight theoretical predictions for its performance plotted in Fig. 1.

### E.1. Specializing $g_{\text{out}}$ to specific activations for multi-layer attention in the $M = 2$, $P_l = 1$ case

We start by rewriting $g_{\text{out}}$ as an integral

$$[g_{\text{out}}(y, \omega, V)]_{lm} = \frac{\int dZ\, e^{-\sum_{l,l',m=1}^{L,2} \frac{(Z_{lm} - \omega_{lm})(Z_{l'm} - \omega_{l'm})}{2V_{ll'}}} \sum_{l'=1}^{L} (Z_{l'm} - \omega_{l'm}) V_{l'l}^{-1} \delta\left(y - B_c^L(Z)\right)}{\int dZ\, e^{-\sum_{l,l',m=1}^{L,2} \frac{(Z_{lm} - \omega_{lm})(Z_{l'm} - \omega_{l'm})}{2V_{ll'}}} \delta\left(y - B_c^L(Z)\right)} \tag{157}$$

We can now massage the integrals in the numerator and denominator. Let us take the denominator as an example. We can

isolate the integral over $Z_{m,L}$

$$\int dZ\, e^{-\sum\limits_{l,l',m=1}^{L,2} \frac{(Z_{lm}-\omega_{lm})(Z_{l'm}-\omega_{l'm})}{2V_{ll'}}} \delta\left(y-B_c^L(Z)\right) \tag{158}$$

$$= \int \left[\prod_{l=1}^{L-1} dZ_l\right] e^{-\sum\limits_{l,l',m=1}^{L-1,2} \frac{(Z_{lm}-\omega_{lm})(Z_{l'm}-\omega_{l'm})}{2V_{ll'}}} \int dZ_L e^{-\sum\limits_{l,m=1}^{L,2} \frac{(Z_{lm}-\omega_{lm})(Z_{Lm}-\omega_{Lm})}{2V_{lL}}} \tag{159}$$

$$\times\, \delta\left(y-\sigma\left(Z_L^\top(B_c^{L-1}(Z_1,...,Z_{L-1})^\top B_c^{L-1}(Z_1,...,Z_{L-1})Z_L)\right)\right) \tag{160}$$

$$= \int \left[\prod_{l=1}^{L-1} dZ_l\right] \frac{e^{-\sum\limits_{l,l',m=1}^{L-1,2} \frac{(Z_{lm}-\omega_{lm})(Z_{l'm}-\omega_{l'm})}{2V_{ll'}}}}{\left|\det B_c^{L-1}(Z_1,...,Z_{L-1})\right|} \int dZ_L e^{-\sum\limits_{l,m=1}^{L,2} \frac{((B_c^{L-1})^{-1}Z_{lm}-\omega_{lm})((B_c^{L-1})^{-1}Z_{Lm}-\omega_{Lm})}{2V_{lL}}} \delta\left(y-\sigma\left(Z_L^\top Z_L\right)\right)$$
$$\tag{161}$$

$$= \int \left[\prod_{l=1}^{L-1} dZ_l\right] \frac{e^{-\sum\limits_{l,l',m=1}^{L-1,2} \frac{(Z_{lm}-\omega_{lm})(Z_{l'm}-\omega_{l'm})}{2V_{ll'}}}}{\left|\det B_c^{L-1}(Z_1,...,Z_{L-1})\right|} I_\sigma \tag{162}$$

Intuitively, we mapped an $L \times M$ dimensional integral with a Dirac delta as an argument into a $(L-1) \times M$ dimensional one, which is much more easily integrated numerically. $I_\sigma$ has to be computed analytically. We give two examples with linear and softmax activation in the following two subsections. A similar manipulation can be realized for the numerator. Putting the two procedures together yields, for the case $L = 2$ discussed in the main text:

**Corollary E.1.** *In the case $M = 2$, $P_1 = P_2 = 1$ and softmax activation, the output filter (12) becomes*

$$[g_{\text{out}}(Y,\omega,V)]_{lm} = \frac{\sum\limits_{Z_2} \int dZ_1 \frac{e^{-\sum\limits_{l,l',m=1}^{2} \frac{((B_c^{L-1})^{-1}Z_{lm}-\omega_{lm})((B_c^{L-1})^{-1}Z_{l'm}-\omega_{l'm})}{2V_{ll'}}}}{|\det B_c^1(Z_1)|} \sum\limits_{l'=1}^{2} ((B_c^{L-1})^{-1}Z_{l'm}-\omega_{l'm})V_{l'l}^{-1}}{\sum\limits_{Z_2} \int dZ_1 \frac{e^{-\sum\limits_{l,l',m=1}^{2} \frac{((B_c^{L-1})^{-1}Z_{lm}-\omega_{lm})((B_c^{L-1})^{-1}Z_{l'm}-\omega_{l'm})}{2V_{ll'}}}}{|\det B_c^1(Z_1)|}} \tag{163}$$

*where the sum over $Z_2$ is intended over the solutions of the two equations $B_c^1(Z_1)Z_2 = \pm\gamma(Y_{1,2}, Y_{2,1})$, with $\gamma(x,y)$ defined as*

$$\gamma(x,y) = \frac{1}{\sqrt{\log((x^{-1}-1)(y^{-1}-1))}} \begin{bmatrix} -\log(x^{-1}-1) \\ \log(y^{-1}-1) \end{bmatrix} \tag{164}$$

### E.1.1. COMPUTING $I_\sigma$ FOR LINEAR ACTIVATION AND $M = 2$

We state again $I_\sigma$ in a compact form

$$I_\sigma = \int dz\, \delta(y-\sigma(zz^\top))F[z] \tag{165}$$

where $F$ is a generic function of $z$. It is convenient to write all the matrices explicitly

$$I_{\text{linear}} = \int dz_1\, dz_2\, \delta\left(\begin{bmatrix} y_{11} & y_{12} \\ y_{21} & y_{22} \end{bmatrix} - \begin{bmatrix} z_1^2 & z_1 z_2 \\ z_1 z_2 & z_2^2 \end{bmatrix}\right) F[z_1, z_2] \tag{166}$$

We can see that $y_{12} = y_{12}$, and also that $y_{11}, y_{22} > 0$. On the other hand, the information on $\{y_{11}, y_{12}, y_{22}\}$ is redundant, as one can use either just $\{y_{11}, y_{12}\}$ or $\{y_{22}, y_{12}\}$. We will choose the former, and write the integral as

$$I_{\text{linear}} = \int dz_1\, dz_2\, \delta(y_{11}-z_1^2)\delta(y_{12}-z_1 z_2)F[z_1, z_2] \tag{167}$$

which we can easily solve obtaining

$$I_{\text{linear}} = \frac{F\left[-\sqrt{y_{11}}, -\frac{y_{12}}{\sqrt{y_{11}}}\right] + F\left[\sqrt{y_{11}}, \frac{y_{12}}{\sqrt{y_{11}}}\right]}{2|y_{11}|} \tag{168}$$

E.1.2. COMPUTING $I_\sigma$ FOR SOFTMAX ACTIVATION AND $M = 2$

Let us state again the problem

$$I_{\text{softmax}} = \int dz_1 \, dz_2 \, \delta \left( \begin{bmatrix} y_{11} & y_{12} \\ y_{21} & y_{22} \end{bmatrix} - \text{softmax} \begin{bmatrix} z_1^2 & z_1 z_2 \\ z_1 z_2 & z_2^2 \end{bmatrix} \right) F[z_1, z_2] \tag{169}$$

This case is more involved. Let us write out explicitly the softmax

$$\text{softmax} \begin{bmatrix} z_1^2 & z_1 z_2 \\ z_1 z_2 & z_2^2 \end{bmatrix} = \begin{bmatrix} \frac{e^{z_1^2}}{e^{z_1^2} + e^{z_1 z_2}} & \frac{e^{z_1 z_2}}{e^{z_1^2} + e^{z_1 z_2}} \\ \frac{e^{z_1 z_2}}{e^{z_2^2} + e^{z_1 z_2}} & \frac{e^{z_2^2}}{e^{z_2^2} + e^{z_1 z_2}} \end{bmatrix} \tag{170}$$

This tell us that $y_{11} + y_{12} = y_{22} + y_{21} = 1$. A less immediate identity is that $y_{12} + y_{21} \leq 1$, as we can see from these simple manipulations

$$y_{12} + y_{21} \leq 1$$
$$\frac{e^{z_1 z_2}}{e^{z_1^2} + e^{z_1 z_2}} + \frac{e^{z_1 z_2}}{e^{z_2^2} + e^{z_1 z_2}} \leq 1 \tag{171}$$
$$e^{2 z_1 z_2} \leq e^{z_1^2 + z_2^2}$$
$$2 z_1 z_2 \leq z_1^2 + z_2^2$$

The last inequality is always true because of the arithmetic and geometric mean inequality. We choose to keep only the information in $\{y_{12}, y_{21}\}$. We will thus get

$$I_{\text{softmax}} = \int dz_1 \, dz_2 \, \delta \left( y_{12} - \frac{e^{z_1 z_2}}{e^{z_1^2} + e^{z_1 z_2}} \right) \delta \left( y_{21} - \frac{e^{z_2 z_2}}{e^{z_1^2} + e^{z_1 z_2}} \right) F[z_1, z_2] \tag{172}$$

Let us try to solve the following system of equations:

$$\begin{cases} y_{12} &= \frac{e^{z_1 z_2}}{e^{z_1^2} + e^{z_1 z_2}} \\ y_{21} &= \frac{e^{z_1 z_2}}{e^{z_2^2} + e^{z_1 z_2}} \end{cases} \tag{173}$$

which can be rewritten as

$$\begin{cases} \log \left( \frac{1}{y_{12}} - 1 \right) &= z_1^2 - z_1 z_2 \\ \log \left( \frac{1}{y_{21}} - 1 \right) &= z_2^2 - z_1 z_2 \end{cases} \tag{174}$$

This is a simple system of two second-degree equations in $z_1$, $z_2$. The solutions are

$$\begin{cases} z_1 &= \pm \frac{\log \left( \frac{1}{y_{12}} - 1 \right)}{\sqrt{\log \left( \frac{1}{y_{12}} - 1 \right) \log \left( \frac{1}{y_{21}} - 1 \right)}} \\ z_2 &= \mp \frac{\log \left( \frac{1}{y_{21}} - 1 \right)}{\sqrt{\log \left( \frac{1}{y_{12}} - 1 \right) \log \left( \frac{1}{y_{21}} - 1 \right)}} \end{cases} \tag{175}$$

This defines a mapping $(y_{12}, y_{21}) \mapsto (z_1, z_2) = \pm\gamma(y_{12}, y_{21})$. The Jacobian of the transformation is

$$|\det J_\gamma| = \frac{1}{2 \left| 2 y_{12} y_{21} (y_{12} - 1)(y_{21} - 1) \log \left( \frac{y_{12} y_{21}}{(y_{12} - 1)(y_{21} - 1)} \right) \right|} \tag{176}$$

Notice that all the quantities above are real-valued because $y_{12} + y_{21} \leq 1$.

Finally, we obtain

$$I_{\text{softmax}} = \frac{F\left[ -\gamma(y_{12}, y_{21}) \right] + F\left[ \gamma(y_{12}, y_{21}) \right]}{2 \left| y_{12} y_{21} (y_{12} - 1)(y_{21} - 1) \log \left( \frac{y_{12} y_{21}}{(y_{12} - 1)(y_{21} - 1)} \right) \right|} \tag{177}$$

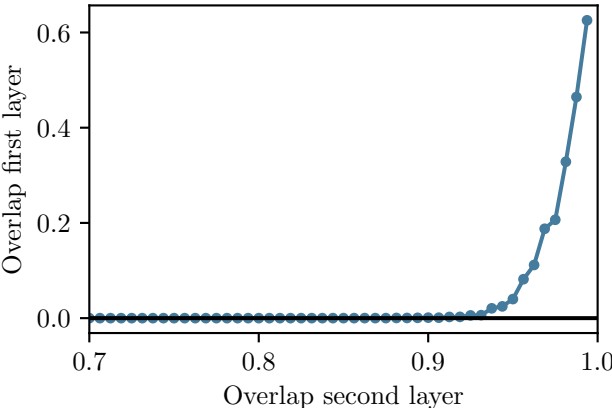

*Figure 5.* **Left**: Overlap of the first layer $Q_{11}$ in two layers of rank one attention with skip connection $c = 1$ and sample complexity $\alpha = 1$ as a function of the overlap of the second layer $Q_{22}$, kept fixed during the iteration. We can see that unless $Q_{22}$ is almost one, $Q_{11}$ is not learned.

## E.2. Weak recovery thresholds for two layers of rank one attention

In this section we describe in detail how to obtain the weak recovery thresholds for the model in Section 3.1. We start by computing $G := \partial_\omega g_{\text{out}}(Y, 0, \mathbb{1}_P)$. We can write it compactly as

$$G_{ll'mm'} = \frac{\int \mathrm{d}Z \, e^{-\frac{1}{2} \sum\limits_{l,l',m=1}^{2,2} Z_{lm} Z_{l'm}} (Z_{lm} Z_{l'm'} - \delta_{ll'}) \delta(y - g(Z))}{\int \mathrm{d}Z \, e^{-\frac{1}{2} \sum\limits_{l,l',m=1}^{2,2} Z_{lm} Z_{l'm}} \delta(y - g(Z))} \tag{178}$$

Since $g$ is an even function, $G_{ll'mm'}$ is zero if $l \neq l'$. This implies that the operator $\mathcal{F}$ in (15) is diagonal, hence $\alpha_1$ is simply

$$\frac{1}{\alpha_1} = \sup_{\mathcal{X} \in \mathbb{S}_+^{P_l}, \|\mathcal{X}\|=1} \|\mathcal{F}(\mathcal{X})\| = \max_l \sum_{m,m'=1}^{2} \mathbb{E}_Y[(G_{llmm'})^2] \tag{179}$$

Implementing the formula above reveals that for these models the maximum is achieved for $l = 2$. Computing $\alpha_2$ is harder, but we can make the simplifying assumption that for $\alpha > \alpha_2$ the second layer is perfectly recovered, so

$$Q = \begin{pmatrix} 0 & 0 \\ 0 & 1 \end{pmatrix} \tag{180}$$

On the other hand, if the second layer is perfectly recovered, one has to simply compute a linear perturbation on $Q$ along $Q_{11}$, giving us

$$\frac{1}{\alpha_2} = \sum_{m,m'=1}^{2} \mathbb{E}_Y[(G_{11mm'})^2] \tag{181}$$

We would like to stress that having learned the second layer is crucial for the learning of the first, and it's central to the grand staircase phenomenon. We can see this also in the model of attention. Suppose we run AMP and choose not to keep the weights in the second layer at a fixed overlap $Q_2 2^*$ with the target: with all other parameters fixed, the value of the overlap of the second layer at convergence depends strongly on $Q_2 2^*$, as shown in Figure 5 for $\alpha = 1$. In this particular case AMP would learn the second layer almost perfectly and achieve a good overlap in the first one, as we can see in Figure 1 in the main. On the other hand, if we prevent the algorithm to learn the second layer, then also the first one is not learned.

## E.3. Language and hardware specific details

Here we discuss the coding details of the implementation of GAMP and the state evolution that we use for the figures. The interested reader can look at the code repository at `https://github.com/SPOC-group/SequenceIndexModels`.

A common ingredient is the denoising function $g_{\text{out}}(Y, \omega, V)$ in (163), which is just a two-dimensional integral over a smooth function. We perform the integral by quadrature using the function `dblquad` in Scipy on the set $[-3, 3] \times [-3, 3]$. Additionally, we regularize the integrand by adding $10^{-3}$ to the diagonal of $V$ and $10^{-6}$ to the square root argument in (163). For GAMP we need also to compute $\partial_\omega g_{\text{out}}$, which we do by centered finite differences computed on two points at distance $10^{-5}$. For state evolution we have to perform and expectation over $Y, \xi$ as in (14). We choose to do so using a Monte Carlo method, that is by taking $1440$ samples of $Y$ and $\xi$ and computing $g_{\text{out}}$ for each of them. At each iteration we symmetrise the overlap $Q$.

For each run of GAMP or the state evolution we use 2 Intel Xeon Platinum 8360Y processors and approximately 290 GB of RAM. In Fig. 1 (left) we smoothen the prediction error and the overlap theory curves by interpolating with splines above the critical threshold.

## F. Details on the experiments on real datasets

In this Appendix, we provide further details on the numerical experiment presented in Fig. 3, namely the training of a simple transformer model on the TREC (Hovy et al., 2001; Li & Roth, 2002) classification task.

**Data –** The TREC dataset (Hovy et al., 2001; Li & Roth, 2002) contains 5500 labeled questions of average length $\approx 10$, divided in 6 classes, classifying the type of the question, e.g. whether it bears on a concept, human, or numerical value. The vocabulary size is 8700. The data is pre-processed using the uncased base BERT model (Kenton & Toutanova, 2019) and padded into sequences of length $M = 32$.

**Architecture –** We consider a small transformer model with two consecutive self-attention layers, with tied key and query matrices, and a fully-connected readout layer. More precisely, the architecture comprises

- **Embedding layer**: embeds the input sequence in dimension $D = 128$

- **Attention layers**: two consecutive self-attention layers

$$\boldsymbol{x}_l = \boldsymbol{x}_{l-1} \operatorname{softmax} \left[ \frac{\boldsymbol{x}_{l-1}^\top \boldsymbol{w}_l \boldsymbol{w}_l^\top \boldsymbol{x}_{l-1}}{D} \right]$$

  with tied weights $\boldsymbol{w}_l \in \mathbb{R}^{D \times P_l}$, and hidden dimension $P_1 = P_2 = 64$.

- **Pooling**: the sequence after the attention layers $\boldsymbol{x}_2 \in \mathbb{R}^{D \times M}$ is averaged over its second dimension.

- **Fully-connected readout:** linear projection from dimension $D$ to 6 (number of classes).

The architecture is then trained over the cross-entropy loss, using the `Pytorch` (Paszke et al., 2017) implementation of the `AdamW` (Loshchilov & Hutter, 2019) optimizer, using learning rate $\eta = 5 \times 10^{-5}$ and batch size 16. The weights $\boldsymbol{w}_l$ at initialization are sampled from a Gaussian distribution with zero mean and standard deviation $10^{-4}$.

