# OpenReview forum: "Fundamental limits of learning in sequence multi-index models and deep attention networks: high-dimensional asymptotics and sharp thresholds"
_ICML.cc/2025/Conference — ICML 2025 poster_

### Official Review · Reviewer_jD8f · 2025-03-11

**Overall Recommendation:** 3

**Summary:**

This paper uses GAMP to quantify the Bayes-optimal performance of deep attention networks. They show that such a deep attention network can be mapped into a sequence multi-index model, which is a variant of the basic multi-index model but applied to a sequence of data of fixed length M.

-- update after rebuttal --

This paper presents an interesting application of AMP to the attention mechanism, so I maintain my evaluation. The authors have addressed most of my questions, although I look forward to future research on the order of learning (e.g. in Bayes-optimal learning or SGD) that this paper identified as an open problem.

**Claims And Evidence:**

The analysis of this paper follows naturally from the existing analysis for multi-index models.

I think it’s important to clarify early on, maybe around (1), that the scaling regime considered in this paper. By “deep”, we refer to a constant number of layers L, rather than infinite. (there is only one short paragraph on $L\to\infty$) The growing quantities are the number of data samples N and the length of the token D.

line 229: my understanding is that AMP (i.e. the fixed point of SE) attains the largest stationary points of the replica potential. The global minimizer of the potential corresponds to the Bayes-optimal performance. Therefore when the potential function has a unique stationary point, then AMP attains the Bayes-optimal performance. Is this correct? If so, I think it would be better to be precise with terminology e.g. stationary points vs global minimiser vs local extremiser.

The paper is heavy in notation, so I think it's important to carefully define every term and its dimension, especially when you present the transformation from the deep attention network to SMI. See Other Comments Or Suggestions for details.

**Essential References Not Discussed:**

NA

**Experimental Designs Or Analyses:**

Could you explain why the off-diagonal entries of Q are zeros? I’d imagine the estimate of a given token should be correlated across different layers of the network?

Figure 1 Right: how did you define the weak recovery threshold? please add legend.

Figure 2: why does the similarity curve start at a nonzero value for GAMP?

**Methods And Evaluation Criteria:**

AMP is a principled way of characterising the Bayes-optimal performance.

What I find confusing though is the order of learning. Do you have any explanation as to why, compared to GAMP or SGD, the order of learning is reversed in your TREC example? ie earlier layers in the network are learned first? I think this is an important behaviour that the paper claimed that AMP should capture correctly, but why is this reversed in practice? Perhaps the method you used for solving the TREC task is far from Bayes-optimal?

**Other Comments Or Suggestions:**

line 20: Bayes-optimal

line 21: what do you mean by “commensurably large”? It might be more accurate to say “proportional”, large dimension D—> token length? Be more precise. “D-dimensional” is cumbersome, could use “length-D” instead

eqns (2)-(3) I’d suggest reverse the order and define x0 first then the recursive update rule for xl, and finally yDA(x). Also, maybe clarify that w_l is the key and query matrix for layer l instead of calling w_l a generic weight matrix. Is the value matrix absorbed into the activation function here? Maybe clarify that.

In (3) $\mathbb{1}_M$ denotes the MxM all-one matrix, instead of the identity right? clarify notation here

line 55: check linespace

line 70: “N samples of Gaussian iid input sequences”, clarify that each sample refers to M sequences of len-D tokens.

line 159: barer input not output right?

line 364: Figures 2 top or bottom instead of left or right

typos in eq (40)

lots of typos around (61)-(62): I thought Bt should be NxPM? instead of the stated DMxPM?

**Other Strengths And Weaknesses:**

Generally well-written, but notation especially those in the supplementary material can be improved.

I think the title is slightly too long.

**Questions For Authors:**

Clarification questions (addressing these will likely improve the paper and change my evaluation of the paper):

1. line 92: B_c^l what does the subscript c stand for?

2. (13)-(14) can you clarify the dimension of the SE iterates? I think they are PxP matrices?

3. (5): what are the deltas here? I didnt get this formula

4. line 218: “with a non-separable prior on the parameters” can you elaborate what exactly is non-separable? Is your denoiser g_out rowwise non-separable?

5. line 357: “Since these first layers are interchangeable we expect that also the overlaps should share the same invariance, making it possible to simplify the state evolution equations in this limit, in a similar spirit to (Aubin et al., 2018).” Do you mean you can simplify the SE equations treating the first L-1 layers essentially as the same so captured by one set of parameters, while the L-th layer itself can be described by a second set of parameters? So overall your SE will only involve (P1+PL)x(P1+PL) matrices instead of PxP matrices, which will now have infinite dim since L\to \infty.

6. Most important question: Do you have any explanation as to why, compared to GAMP or SGD, the order of learning is reversed in your TREC example?

7. The analysis of this paper seems to follow naturally from the existing analysis for multi-index models, in the sense that the SE is dim-free so extending from M=1 to arbitrary M doesnt really change the SE equations. Can you elaborate what are the main technical challenges that you've resolved?

8. $g: \mathbb{R}^{P\times M}\to \mathbb{R}^K$. What does the K-dimensional output represent? It might worth adding one sentence giving an example. I'm guessing this could represent say the sentiment of the input?

9. I'm confused by your application of the non-separability result from Gerbelot & Berthier 2023. GB23 allows non-separability across the N data samples, whereas it seems that in (61)-(62) the non-separability in your work is within each sample as you treat the N samples as iid. By a "sample" I mean M length-D tokens. Can you clarify? Can you also clarify whether Bt should be NxPM in (61)-(62)?

10. Is it possible to incorporate the feedforward layers alongside the attention mechanisms?

**Relation To Broader Scientific Literature:**

Interesting paper that might help our understanding of the learning dynamics in transformers.

**Theoretical Claims:**

I didn't check the proofs in detail but believe they followed naturally from the existing analysis for multi-index models.

---

> ### Author Rebuttal · Authors · 2025-03-31
>
> We thank the reviewer for their constructive comments. Their remarks will allow us to improve the clarity of our manuscript greatly.
>
> **Claims And Evidence**
>
> Clarifying the limit: By "deep," we mean constant ($O(1)$) depth. Our results hold in the large $D$, proportionally large $N$ limit.
>
> Stationary points vs. minimizers: We agree and will refine the mapping between SE fixed points and the free energy around line 229 (see also response to Reviewer xpYy).
>
> **Methods And Evaluation Criteria**
>
> Order of learning: The reversed learning order is not specific to the algorithm. On our synthetic task, all algorithms learned the last layer first. In TREC, the reversal likely stems from task, model, and algorithm jointly. Our theory predicts that Bayes-optimal learning proceeds layer-wise from the last layer. While this matches some empirical observations, the learning order depends intricately on task and architecture and deserves further study. We will note this.
>
>
> **Experimental Designs Or Analyses**
>
> The diagonal structure of $Q$ is task-specific. Iterating SE from generic $Q$ shows it remains diagonal. Intuitively, the target weights are random and approximately orthogonal in high dimensions, implying the same for the GAMP estimates.
>
> *Fig. 1 (right): define weak recovery threshold.*
> Defined in Theorem 2.3: the smallest sample complexity at which GAMP’s estimates have non-zero correlation with the target. We will add a pointer in the caption.
>
> *Fig. 2: nonzero start* The first point shows overlap after one iteration—not initialization. We will clarify.
>
> **Supplementary Material**
>
> Thank you for the note, we will improve that appendix to be better self-contained.
>
> **Other Comments Or Suggestions**
>
> *line 21*  We indeed mean "proportionally large", we will reword.
>
> *Suggestions on eqns (2)-(3)*: Thank you, we will follow your suggestion for clarification.     As we discuss in l-157, we consider frozen value matrices fixed to identity. We will stress this point.
>
> *In (3) $1_M$ denotes*: In (3) $1_M$ refers to the identity matrix.
>
> *barer input and other typos*: We thank the reviewer for their careful reading. These typos will be corrected.
>
> **Questions For Authors**
> 1. c is the skip connection strength, introduced in (3).
>
> 2. They are indeed PxP matrices. We will stress this point.
>
> 3. They are Kroneker deltas: $\delta_{ab} = 1$ if $a=b$ and zero otherwise.
>
> 4. We mean that the prior on the weights $W^*$ is nonseparable. The denoiser $g_{out}$ is defined in (12) and its functional form can be arbitrarily complicated, both column-wise and row-wise non-separable, as  in the case of deep attention models, as detailed in Appendix E.
>
> 5. We mean that it would be natural to expect we need less order parameters: in general one would need $PL \times PL$, but if the first $L$ layers are identical we just need $4P \times 4P$ by imposing that the overlap $Q$ has a certain structure. More in detail, we would be imposing that the overlap of a layer $l$ and the $L-$th layer is identical for any $l\le L-1$, that the overlap between any pair of layers $l_1,l_2\le L-1$ is identical, and finally that the self-overlap of the weights of any layer $l\le L-1$ is identical.
>
> 6. We refer to the main answer above to this same question.
>
> 7.  The SE equations do involve the dimension $M$ through the denoiser $g_{\rm out}: \mathbb{R}^{M\times M} \to \mathbb{R}^{P\times M}$, involved in the SE equation (14), and defined in (12). A technical challenge was, in fact, precisely to ascertain the denoiser $g_{\rm out}$, which we unravel in Appendix E. Another challenge was to actually show the equivalence between SMIMs and MIMs (see appendix A).
>
>
> 8. $K$ denotes the output dimension—for sentiment classification, this corresponds to the predicted class probabilities. In our setting, the output is an $M \times M$ matrix, so $K = M^2$, interpretable as a target attention matrix capturing token interactions. Sentiment analysis, with $K$ being the number of sentiments, is a good example that we will include.
>
> 9. The setup in GB23 and BMN20 (Berthier, Montanari, Nguyen, 2020) for non-symmetric matrices allows for non-separable non-linearities along both the axis i.e across different $N$ samples and across different coordinates of the parameters. See for instance equation $(5)$ in  GB23 and equations $(1),(2)$ in BMN20. In Equations (61)-(62), $B^{t}$ contains the flattened parameters $\hat{W}^t$ and therefore has dimensions $DM \times PM$. The number of samples $N$ appears in the dimensions of $\Omega^t \in \mathbb{R}^{N \times PM}$ which contains the corresponding flattened inputs.
>
> 10. Adding fully trainable non-linear layers is a challenging pursuit which would warrant completely new technical ideas, and we defer the study thereof to future work. Non-learnable non-linear layers could be added without a big additional technical challenge.

---

> > ### Comment · Reviewer_jD8f · 2025-04-01
> >
> > I thank the authors for their point-to-point response, and decide to keep my score.

---

### Official Review · Reviewer_R4QM · 2025-03-11

**Overall Recommendation:** 3

**Summary:**

This paper extends the theoretical framework of multi-index models to sequence models and derives a sharp asymptotic characterization of the optimal performance of the generalized approximate message-passing (GAMP) algorithm. This paper also characterizes sharp thresholds on the minimal sample complexity required for better-than-random prediction performance.

**Claims And Evidence:**

Yes.

**Essential References Not Discussed:**

No.

**Experimental Designs Or Analyses:**

I check the soundness of the experimental designs, which makes sense for me.

**Methods And Evaluation Criteria:**

Yes.

**Other Comments Or Suggestions:**

No.

**Other Strengths And Weaknesses:**

Strengths:
1. This paper is well organized and easy to follow.
2. The experiments are clear.

**Questions For Authors:**

Questions:
1. This paper considers the best-known polynomial-time algorithm (AMP) algorithm. I am wondering if it is possible to get similar theoretical results, if we change the AMP algorithm to gradient descent algorithms.

**Relation To Broader Scientific Literature:**

The optimal performance of gradient descent algorithms has been well studied for multi-index models, the key contributions of the paper is   that it extends the theoretical analysis of multi-index models to sequence multi-index (SMI) functions.

**Theoretical Claims:**

No.

---

> ### Author Rebuttal · Authors · 2025-03-31
>
> We thank the reviewer for their constructive comments.
> Numerically, we observe that in the setting of Fig. 2, GAMP and  Gradient Descent  display similar layer-wise learning dynamics. To put these observation on completely rigorous theoretical grounds, one would need to adapt the dynamical-mean-field-theory analysis of (Gerbelot, Troiani, Mignacco, Krzakala, Zdeborová 2024) to SMIMs. While we believe such an extension to be possible, it would warrant significant additional investigation, and we leave this for future work.

---

### Official Review · Reviewer_KcvH · 2025-03-12

**Overall Recommendation:** 3

**Summary:**

This paper studies the fundamental limits of learning in deep attention networks by establishing a connection to sequence multi-index models. The key contributions include the mapping from deep attention networks to SMI models and theoretical characterization of statistical and computational limits of the models using AMPs.

**Claims And Evidence:**

The claim that deep attention networks map to SMI models is well verified through formal derivations (Section 2, Appendix A), and the claim AMP achieves Bayes-optimal performance is also demonstrated through both theoretical analysis (Lemma 2.2, Theorem 2.3) and synthetic experiments (Fig. 1–2).

**Essential References Not Discussed:**

As far as I know, this work has a comprehensive discussion on literatures. Most of the related literatures that I know are well addressed.

**Experimental Designs Or Analyses:**

The synthetic experiments clearly show phase transitions and layer-wise learning. However, it would be better if there are some real-world evaluations.

**Methods And Evaluation Criteria:**

The theoretical analysis leverages AMP and state evolution, and the synthetic experiments are well-controlled (Gaussian inputs, large dimensions) and verify the performance of AMP. However, the real-world evaluation is limited, since the experiment is based on single dataset and simplified architecture.

**Other Comments Or Suggestions:**

Minor typos: Page 3, "idependently" → "independently"

**Other Strengths And Weaknesses:**

This work has novel theoretical connection between attention networks and SMIs with rigorous phase transition analysis. However, it needs more real-world evaluation to address its practical implications.

**Questions For Authors:**

I have no other questions.

**Relation To Broader Scientific Literature:**

This work connects to multi-index models and extends single-layer attention analyses, which is highly relavent and important to the area of feature learning theory.

**Theoretical Claims:**

Theorems 2.1 and 2.3 rely on mapping to multi-index models and prior AMP analyses. Proofs in appendices seem rigorous.

---

> ### Author Rebuttal · Authors · 2025-03-31
>
> We thank the reviewer for their constructive comments. The focus of this work is indeed theoretical, and lies in making several theoretical findings on learning in attention models, extending previous related works. We discussed how the uncovered phenomena also hold with Gradient Descent (Figure 2 Bottom) and on some simple real world case (Figure 3 Right), but totally agree that it would be interesting to have a more thorough mechanistic study of layer-wise learning in more involved transformer models. On the other hand, we believe such an analysis is outside the scope of the paper at hand, and hope our work will inspire a more applied set of researchers to conduct such an investigation.

---

### Official Review · Reviewer_xpYy · 2025-03-14

**Overall Recommendation:** 3

**Summary:**

This paper studied the problem of learning sequential multi-index models. The authors showed a deep connection between the SMI model and deep attention networks, in that the deep attention network can be formulated in the form of a SMI function. The authors studied the limit thresholds of the sample complexity for weak recovery of the hidden directions $W^*$. Numerical experiments are conducted, providing intuitions on the dynamics of the GAMP and SGD algorithms when learning deep attention networks. In particular, they showcased the sample complexity thresholds of which 2 layers of the weights are learned in a hierarchical manner.

**Claims And Evidence:**

Almost. There is a conjecture about the learning dynamics of 3 layer attention net, supported by some numerical experiments of a special case.

**Essential References Not Discussed:**

No.

**Experimental Designs Or Analyses:**

yes. no significant issues.

**Methods And Evaluation Criteria:**

Theory paper, N/A.

**Other Comments Or Suggestions:**

See weakness.

**Other Strengths And Weaknesses:**

Weakness:
1. I am not sure what conclusions we should draw from this set of results. The authors proposed a bayes-opitmal GAMP method for learning attention networks and SMIs, where it needs calculations of the posterior distribution $E[Z|Y]$, where $Y$ are the labels. The choice of this GAMP method makes sense somehow since this is one of the optimal first-order methods. However, it is not clear it is optimal in what sense. Is this optimal in the sense of sample complexity, or it is optimal in terms of runtime as well? If the algorithm achieves optimal sample complexity but have exponential runtime, then this sample complexity threshold seems not very informative. Or, are the authors trying to provide a kind of lower bound, since the GAMP is 'optimal' first-order algorithm?
2. I am afraid this paper is not very clear written. It is particularly confusing to me the phrase in Theorem 2.3, what are the differences between 'weak recovery for threshold for initial recovery' and the 'subspace recovery threshold'? What are the differences between these notions and the notion of weak subspace recovery in prior works, like definition 2 in [1]?
3. Continuing with the second point above, I am very confused about why there should be 2 different thresholds. Is this coming from the theory that there are, say, some local minima or some unstable fixed point that the power iteration cannot overcome?
4. What's more, there seems to be 2 Q^t's, coming from (14) and (18), which are very confusing to me. Are they measuring the same 'non-vanishing of overlaping' between $W^t$ and $W^*$? Perhaps I am wrong but my understanding about this thread of research it to show that after some threshold $\alpha$, the fixed point of the AMP process/power iteration Q would be p.d. and is a fixed point. Therefore, when the authors talk about fixed points and weak recovery thresholds (in particular in (16)), which process are the authors referring to?
5. The authors observed that the second threshold alpha_1 is the sample complexity threshold for learning the first layer weights after learning the second layer. Are there any theoretical justifications?
6. Finally, I am confused about the difference between SMI and MIM. As the authors mentioned, the SMIs can be essentially viewed as a MIN (in (49)), in particular the new MIM has hidden dimension $PM = O(1)$, then why we need a new set results on SMIs?

**Questions For Authors:**

When $P = M = 1$, (i.e., considering SIMs), how are the results relates to the sample complexity of efficient SQ algorithms, i.e.,  the generative exponent?

**Relation To Broader Scientific Literature:**

The key contributions align with a thread of work on the limits of weak recovery of multi-index models using GAMP. The techniques and results are similar to prior works like [1], though I think it needs some more careful analysis with respect to the new structure of the model.

[1] Fundamental computational limits of weak learnability in high-dimensional multi-index models

**Theoretical Claims:**

I skimmed through the proofs but unfortunately I did not check it line by line.

---

> ### Author Rebuttal · Authors · 2025-03-31
>
> We thank the reviewer for their constructive comments.
>
> **Claims And Evidence:**
>
> Fig. 3 numerically supports the conjecture, while Sec. 3.2 discusses its infinite-depth limit where inner layers ($\ell \leq L - 1$) become interchangeable.
>
> **Relation To ... Literature:**
>
> The reviewer is right that our work builds on prior techniques. However, as noted, the new structure introduces substantial technical challenges.
>
> **Weakness**
>
> 1.  On optimality:
>
> (1) GAMP is a polynomial-time algorithm that converges with few iterations, as predicted by state evolution. All examples in our plots used at most 50 iterations—we will state this explicitly.
>
> (2) First-order optimality: GAMP minimizes prediction error among all first-order algorithms (e.g., Celentano et al. 2020), making our thresholds optimal in that class.
>
> (3) Bayes-optimality: Our Th. 2.1 shows that when state evolution has a unique fixed point, it corresponds to the MMSE estimator.  In such cases, GAMP achieves statistical optimality and our thresholds match the fundamental limits.  This holds in all our examples.
>
> 2. Thank you—this helps clarify terminology. The weak recovery threshold for initial recovery is the sample complexity beyond which $\hat{W}$ escapes the zero fixed point, i.e., gains non-zero row-space. In contrast, the subspace recovery threshold for a subspace $U$ of the target row-space is the sample complexity above which $U$ is included in the row-space of $\hat{W}$.
> Def. (17), equivalent to Def. 2 in [1], formalizes this:
> $v^\top Q_t v > 0$ ensures $v \in U$ is in $\hat{W}W^\star$'s row space, where $Q_t = \hat{W}W^{\star \top}/d = \hat{W} \hat{W}^\top/d$.
>
> For SMI targets, GAMP dynamics are more complex than for single-index: Learning is coupled across subspaces—progress on one can aid others. This manifests as GAMP passing near a series of saddle points. The thresholds in (17) reflect the minimal sample complexities required to escape these saddles, matching those in [1] (Def. 5), found via the stability of (18).
>
> The threshold in (16) is the minimal complexity to recover any 1D subspace, thus a special case of the more general subspace recovery threshold
>
> 3. The weak recovery threshold for initial recovery marks the point where there exists a subspace $U$ that satisfies the condition (17). For example, in sec. 3.1, span$(w^\star_2)$ is recovered first, so its subspace recovery threshold (17) coincides with the weak recovery threshold for initial recovery (16). In contrast, the threshold for span$(w^\star_1)$ is higher, and depends on how much of $w^\star_2$ has been learned. It is not equal to (16): One can define a more precise threshold $\alpha_{w^\star_1}(q_2)$—the sample complexity for recovering span$(w^\star_1)$ given overlap $q_2$ with $w^\star_2$, assuming minimal $q_2$ in (17). Thus, (17) gives a fine-grained notion of threshold for a specific subspace, while (16) captures the threshold for recovering any subspace.
>
> 4. In (18), $Q_t$ denotes the overlap between $\hat{W}^t$ and $W^\star$ when the estimator uses side-information of the form $\lambda W^\star + \sqrt{1 - \lambda} \times \text{noise}$ (see l. 254, right column). Eq. (14) corresponds to the case $\lambda = 0$, with no side-information beyond the dataset ${x^\mu, y(x^\mu)}$. Thus, (14) and (18) describe distinct AMP algorithms, depending on the presence of side-information. Introducing side-information is technically required to define the thresholds (17) via the limit $\lambda \to 0$. For clarity, we will denote the overlap as $Q^t_\lambda$ in Theorem 2.3.
>
> 5. The thresholds in Fig. 1 indeed instantiate the theory of Sec. 2.
> We will revise $\alpha_1$ in (17) to $\alpha_U$.
> In subsection 3.1, $\alpha_1=\alpha_{{\rm span} (w^\star_2)}$, i.e. the sample complexity necessary to retrieve the subspace spanned by the second layer weights, is the weak recovery threshold from (16). The sample complexity $\alpha_2$ corresponds to the sample complexity required to further retrieve the full row-space $ {\rm span} (w^\star_2, w^\star_1)$.
> This corresponds to (17) for $U= {\rm span} (w^\star_2, w^\star_1)$.
>
> 6. Writing a MIM as an SMIM naturally leads to studying targets where each input is a set of vectors—a common setup in sequence models. SMIMs can be seen as MIMs with a block-structured weight matrix (see Appendix B). While our proofs build on MIM results, the specific block structure we consider is new and well-suited to attention models. SMIMs are more natural for gradient-based training, where gradients are computed for all weights. The equivalent MIMs involve tied/zeroed weights, incompatible with standard gradient descent. This is similar to how CNNs are special cases of FCNs, but still distinct in practice due to gradient computation.
>
> **Questions For Authors**
>
> We expect GAMP to saturate the SQ bounds, as shown for MIMs in [1] (Lemmas 3.3 and 4.3).
> Weak recovery thresholds relate to generative exponents: 0 if threshold is zero, 1 if finite, $\geq 2$ if infinite.

---

### Decision · Program_Chairs · 2025-05-01

**Decision:**

Accept (poster)

**Comment:**

This paper considers learning of attention networks with O(1) depth, with tied and low-rank weights. Authors map such models to sequence multi-index models, and conduct their analysis on under this relationship. Authors study weak recovery and the sample complexity thresholds. They further provide numerical results, on GAMP and SGD algorithms when learning the multi-index model.

This paper was reviewed by four expert reviewers and received the following Scores: 4x Weak Accept. I think paper is studying an interesting topic but authors are not able to convince the reviewers sufficiently well. The following concerns were brought up by the reviewers:

- The paper and its motivation is not clear.

- Results not sufficiently explained in paper.

No reviewers championed the paper and they are not particularly excited about the paper.
I believe majority of the concerns can be addressed. As such, based on the reviewers' suggestion, as well as my own assessment of the paper, I recommend including this paper to the ICML 2025 program.